# A compendium of Amplification-Related Gain Of Sensitivity genes in human cancer

Veronica Rendo [1,2,3,4,5,17] ✉, Michael Schubert [6,7,8,9,17] ✉,
Nicholas Khuu [1,2,4], Maria F. Suarez Peredo Rodriguez[7], Declan Whyte[7],
Xiao Ling[7], Anouk van den Brink [7], Kaimeng Huang [4,10], Michelle Swift [10],
Yizhou He [4,10], Johanna Zerbib[11], Ross Smith [5], Jonne Raaijmakers[6],
Pratiti Bandopadhayay [3,4,12], Lillian M. Guenther [13], Justin H. Hwang[14],
Amanda Iniguez[15], Susan Moody[1,3,4], Ji-Heui Seo[1], Elizabeth H. Stover [1,3,4],
Levi Garraway[1,4], William C. Hahn [1,3,4], Kimberly Stegmaier [3,4,12],
René H. Medema [6], Dipanjan Chowdhury [4,10], Maria Colomé-Tatché[8,16],
Uri Ben-David [11,18] ✉, Rameen Beroukhim [1,2,3,4,18] ✉ & Floris Foijer [7,18] ✉

While the effect of amplification-induced oncogene expression in cancer is known, the impact of copy-number gains on "bystander" genes is less understood. We create a comprehensive map of dosage compensation in cancer by integrating expression and copy number profiles from over 8000 tumors in The Cancer Genome Atlas and cell lines from the Cancer Cell Line Encyclopedia. Additionally, we analyze 17 cancer open reading frame screens to identify genes toxic to cancer cells when overexpressed. Combining these approaches, we propose a class of 'Amplification-Related Gain Of Sensitivity' (ARGOS) genes located in commonly amplified regions, yet expressed at lower levels than expected by their copy number, and toxic when overexpressed. We validate RBM14 as an ARGOS gene in lung and breast cancer cells, and suggest a toxicity mechanism involving altered DNA damage response and STING signaling. We additionally observe increased patient survival in a radiation-treated cancer cohort with RBM14 amplification.

Due to genomic instability, human cancers accumulate somatic mutations over time. The most frequent type of genomic alterations are copy number changes, affecting on average ~30% of a tumor's genome[1,2]. Somatic copy number alterations (sCNAs) can target focal regions of the genome (e.g., amplification of the oncogene *MYC* in chromosome 8q or deletion of the tumor suppressor *RB1* in chromosome 13q), but often comprise chromosome arm-level events that span hundreds of collaterally-altered genes located in proximity to the cancer driver genes. Such large-scale events shape the transcriptional and translational profile of tumor cells, as chromosomal copy number changes may affect genes with essential roles in tumor progression and viability[3].

In human cancers, changes in DNA copy number tend to be tightly correlated with mRNA expression levels[4,5]. However, uncoupling of gene expression from DNA copy number has been described by multiple mechanisms at the genomic (e.g., rearrangements), epigenetic (e.g., promoter hypermethylation), and post-translational (e.g., buffering copy number imbalances in protein complex members) levels[6–12]. These discrepancies between gene expression levels and copy number suggest that such large-scale changes may have a negative impact on cellular fitness. Such alterations may also create new dependencies. For example, loss of 'Copy number alterations Yielding Cancer Liabilities Owing to Partial losS' (CYCLOPS) genes renders cells dependent on the remaining copy[13]. Similarly, loss of heterozygosity

A full list of affiliations appears at the end of the paper. ✉e-mail: veronica.rendo@igp.uu.se; m.schubert@nki.nl; ubendavid@tauex.tau.ac.il; rameen_beroukhim@dfci.harvard.edu; f.foijer@umcg.nl

events spanning passenger metabolic and essential genes, as well as homozygous losses of gene paralogs, create unique dependencies in tumor cells that can be exploited therapeutically[14–19]. More recently, some studies have also addressed overexpression toxicity of a limited number of genes[20–22], as well as genetic dependencies correlated with chromosome gains in cell lines[12,23,24]. However, the impact of copy number gains affecting "bystander" genes, often co-amplified with oncogenes, remains less well understood.

In this study, we investigate whether copy number gains can also become collateral cancer liabilities, as they affect the expression of multiple genes with diverse biological functions and impact cellular fitness. For this, we hypothesize that some genes located in commonly amplified regions of the genome could be detrimental to the cell when overexpressed upon gain, triggering mechanisms of gene compensation. We identify these 'Amplification-Related Gain Of Sensitivity' (ARGOS) genes by analyzing their gene expression compensation in tumors and cell lines, and their overexpression toxicity in open reading frame (ORF) screens. We experimentally show that the gain of one of these genes, *RBM14*, indeed perturbs the DNA damage response

and cGAS/STING signaling, and its amplification is associated with increased vulnerability of tumors to radiation treatment in a clinical colorectal cancer cohort.

## Results

### Copy-number changes often affect non-driver genes whose expression is nevertheless altered

Much of the genome is frequently gained or lost in cancer (Fig. 1a). This is because individual copy-number alterations, selected by cancer driver events, typically affect large sections of the genome (Fig. 1b). Although Oncogenes (OGs) are more frequently amplified and less frequently lost, and Tumor Suppressor genes (TSGs) show the opposite trend (Supplementary Fig. S1a, b), most of the frequently amplified genes are neither OGs nor TSGs, but likely "collaterally altered" by the gain/loss of large chromosomal regions (Supplementary Fig. S1c, d).

Nevertheless, the expression levels of the vast majority of genes, including these "bystander" genes, scale with their copy number[4,25]. This is true across human tumors[4], across cell lines in the Cancer Cell Line Encyclopedia (CCLE) (Fig. 1c), and also in isogenic RPE-1 cells with individual gained chromosomes[26]. However, we also observe

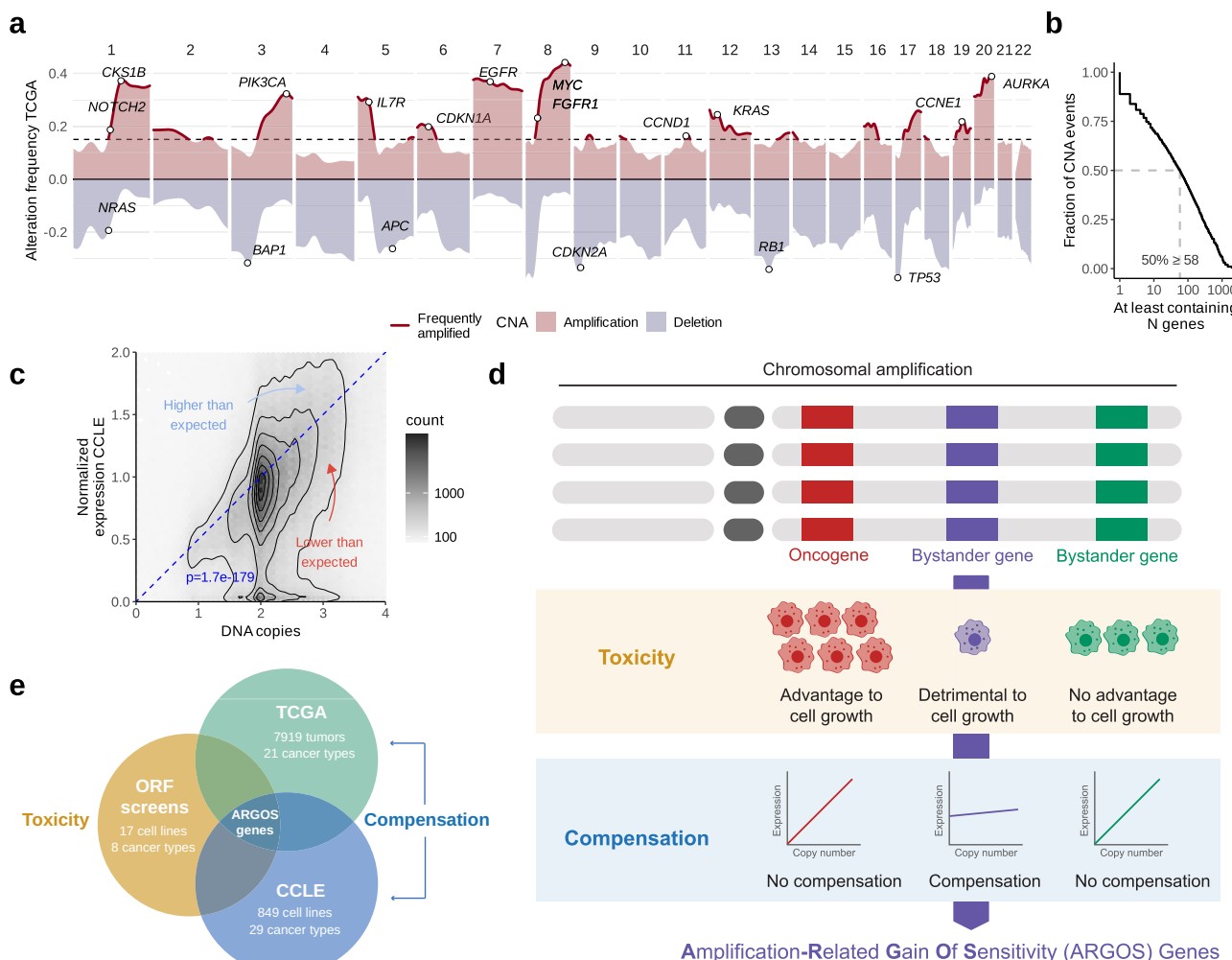

**Fig. 1 | Approach to identify ARGOS genes. a** The landscape of CNAs with the frequency of amplifications and deletions across TCGA tumors shows preferential gains of OGs and losses of TSGs. Genes gained in over 15% of samples (dotted line) were considered commonly amplified. **b** Individual CNA events typically contain multiple genes at a median of 58. **c** Normalized RNA expression across all genes and cell lines scales with DNA copy number in the CCLE, although there is a considerable spread around this trend (scaling *P*-value from linear regression model without intercept). We then statistically model which genes are expressed

consistently lower or higher than the expectation and refer to them as compensated and hyperactivated, respectively. **d** ARGOS genes are identified by genes that are collaterally affected by amplifications, which are also detrimental to cell growth when overexpressed and show compensation upon copy number gain. **e** In practical terms, we employ the CCLE and TCGA cohorts to identify compensated genes and confirm the toxicity phenotype by repurposing previously published ORF screens.

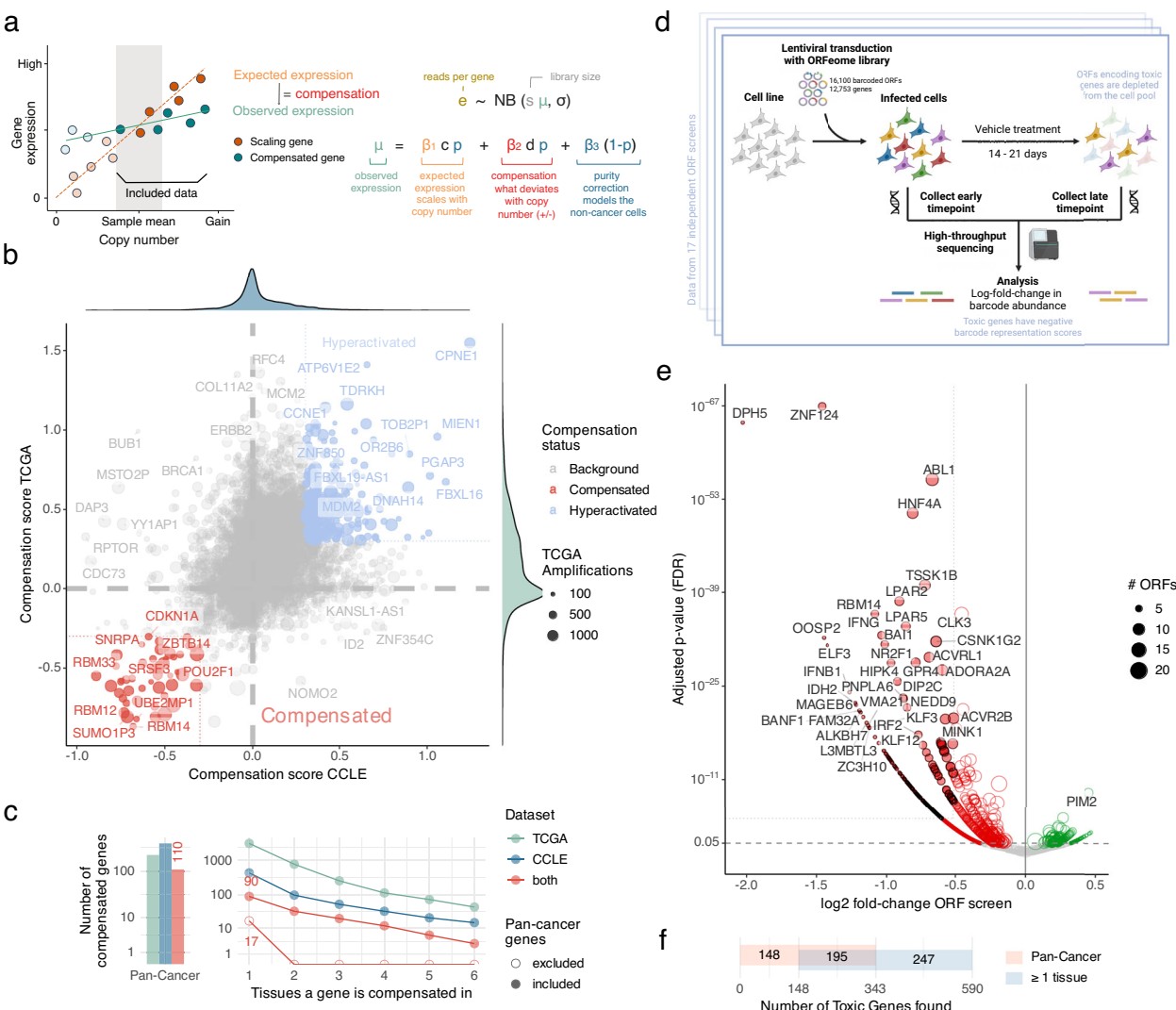

**Fig. 2 | Compensated and toxic genes. a** Using a Bayesian Negative Binomial regression model, we split the expression of each gene in CCLE and TCGA into components that are scaling (orange) and deviating (red) from DNA content. For TCGA data, we explicitly take into account non-cancer cells (blue). We apply this model across (pan-cancer) and for individual cancer types (tissue-specific). **b** In the pan-cancer model, deregulation is well correlated between CCLE and TCGA, and we identify commonly compensated (red) and hyperactivated genes (blue). These are genes that show a compensation score in both data sets of less than − 0.3 or more than 0.3, respectively (dotted lines). **c** Number of pan-cancer (left) and tissue-specific (right) compensated genes, specific to either CCLE or TCGA datasets or common to both. Numbers mentioned in the text are highlighted. **d** We utilize ORF overexpression screens to quantify barcode abundance for outgrowth after the selection marker, which (**e**) identifies genes that are promoting (green) or attenuating (red) cell growth when overexpressed. Genes passing the significance threshold of the linear regression model are shown with a black outline. **f** Numbers of pan-cancer vs. tissue-specific toxic (attenuating) genes and their overlap. Figure 2d was created in BioRender. Beroukhim, R. (2024) https://BioRender.com/z69q647.

---

considerable variation around this trend (Fig. 1c). If amplified genes are consistently expressed at lower or higher levels than expected, we refer to them as "compensated" or "hyperactivated", respectively. We were especially interested in compensated genes because these might reflect negative selective pressures resulting from fitness decreases due to amplification-driven overexpression. That is, genes whose overexpression is "toxic" to cancer cells might exhibit substantial compensation when amplified.

To more directly assess gene toxicity, we assembled gene sensitivity data from 17 different Open Reading Frame (ORF) overexpression screens performed across 8 tumor types[27–35]. We then quantified which genes strongly decreased in abundance across these screens without additional selection pressure. Combining these toxic genes with genes that we identified as "compensated", we aimed to identify the set of ARGOS genes whose amplification could jeopardize cancer cell fitness (Fig. 1d, e).

## Genes are consistently compensated in CCLE and TCGA

We developed a computational method to detect genes that are consistently compensated in their expression relative to their copy number, both across (pan-cancer) and for individual (tissue-specific) cancer types. First, we selected samples with copy-neutral and amplified genes from both large human cancer cell lines (Cancer Cell Line Encyclopedia; CCLE[36]) and tumor (The Cancer Genome Atlas; TCGA[37]) cohorts. We then built a Bayesian Negative Binomial regression model between this copy number and a gene's expression, including a variable for copy number vs. gene expression scaling and one for its deviation (Fig. 2a). In addition, in human tumors, non-cancer cells can represent a large fraction of the cells in a sample (reflecting low tumor purity). These impurities would be expected to modify observed expression levels relative to the expression levels within the cancer cells. To account for this, we explicitly modeled cancer and non-cancer contributions to the observed gene expression in TCGA.

This regression analysis provided us with a compensation score where −1 indicates complete compensation for cancer cells (i.e., no expression changes with amplifications) and +1 indicates full hyperactivation (i.e., twice the gene expression that we would expect based on its purity-corrected copy number change). We built this model across cancers and for individual cancer types (Supplementary Data 1–3).

As expected, most genes scaled with copy number in both the CCLE and TCGA (pan-cancer analysis; Fig. 2b). Compensation scores were significantly correlated between the two datasets, especially when explicitly controlling for non-cancer cells in TCGA data ($P < 10^{-300}$, $R^2 = 0.09$; Supplementary Fig. S2a, b). We did not observe an overall difference in compensation scores between genes that are commonly amplified, deleted, or copy-neutral (Supplementary Fig. S2c); or for OGs, TSGs, and genes that are neither (Supplementary Fig. S2d). However, we found splicing and RNA processing genes (Gene Ontology) consistently compensated over the expected scaling, whereas DNA replication and repair genes were hyperactivated when amplified (Supplementary Fig. S2b).

Using a cutoff of 30% less expression than expected, we identified 110 genes to be compensated across cancer types in both the CCLE and TCGA (Fig. 2b, c and Supplementary Data 1). They contained more protein-coding and fewer non-coding genes (i.e., pseudogenes, lncRNAs) than expected by chance (Supplementary Fig. S2e), and overall much fewer genes than previously reported (Supplementary Fig. S2f). Among the protein-coding genes, we found nine members of the hnRNP family (heterogeneous ribonucleoprotein particle), seven RPLs (Ribosomal protein L), five RBMs (RNA-binding motif), four SRSFs (arginine/serine-rich splicing factor), and three ZNF (Zinc finger) genes. Five are listed as OGs in the COSMIC database[38] (*CHD4, DGCR8, EWSR1, HNRNPA2B1, SRSF3*), four as TSGs (*CHD2, CTCF, FUS, SFQP*), and two are labeled as both (*CDKN1A* and *DDB2*). Given the frequent presence of RNA binding proteins, we performed a separate analysis of genes that contain an RNA recognition motif (RRM) and, more specifically an aggregation-prone disordered region (Prion-Like Domain, PLD[39]). We indeed found a strong enrichment of compensation in RRM-containing genes over protein-coding genes (30x, $P = 10^{-27}$, Fisher's Exact Test) and a further enrichment (10x, $P = 0.0002$) of PLD-containing over RRM genes (Supplementary Fig. S2e).

To validate these as compensated genes, we first examined whether they showed evidence of negative selection in the TCGA. We indeed found higher mutation rates compared to non-compensated genes, consistent with previous studies (Supplementary Fig. S2g). We then compared gene expression changes in isogenic RPE-1 clones with gained chromosomes (either chromosome 7 or a combination of chromosomes 7 and 22; or 8, 9, and 18; for 7, 10, and 7 compensated genes in Supplementary Fig. S2h)[12,23]. We confirmed that the compensated genes residing on the respective gained chromosomes were expressed less than non-compensated genes thereon ($P < 0.05$). We could not confirm this compensation for previously published gene sets (Supplementary Fig. S2i). Furthermore, our compensated genes showed significantly higher associations between genomic gain and diseases other than cancer compared to non-compensated genes, also when compared to other studies ("Triplosensitivity"; Supplementary Fig. S2j)[40]. Combining the evidence of these four independent data sets (TCGA, CCLE, RPE-1 and Triplosensitivity), we therefore consider these 110 genes as having strong evidence of compensation across cancer types and an improvement over previous studies.

In the tissue-specific analysis, we identified a total of 17 additional genes that were compensated in both CCLE and TCGA for at least one cancer type (Fig. 2c). This brings the total number of identified genes common to both datasets to 127, with 90 genes in both the pan-cancer and tissue-specific analysis. The additional tissue-specific genes occurred in one cancer type exclusively. Across pan-cancer and tissue-specific analyses, only a part of the most compensated genes was shared between cell lines (CCLE) and tumors

(TCGA), where the former showed a stronger enrichment in cell cycle and the latter a stronger enrichment in immune-related processes (Supplementary Fig. S2k, l).

## ORF screens reveal genes that are toxic when overexpressed

Dosage compensation could be the consequence of selective pressure to avoid the detrimental overexpression of some genes. To identify genes whose overexpression is indeed toxic, we aggregated gene cell viability data from 17 different ORF screens across 8 tumor types[27–35] (Supplementary Data 1, 4). In each of these ORF screens, cells were transduced with the lentiviral ORFeome library[41] containing 16,100 barcoded constructs encoding for a total of 12,753 genes. After transduction and construct marker selection, cells were subjected to drug treatment or vehicle control and grown for up to 3 weeks. With only the vehicle control arms, we determined the effect of each gene's overexpression on cell viability and/or proliferation. For this, we quantified the $log_2$ fold changes of lentiviral barcodes between early and late time points across all screens using pan-cancer and tissue-specific linear models (Fig. 2d and Supplementary Fig. S3a, b).

Across all cancer cell lines, we observed that many more genes were depleted rather than enriched in these screens, with *DPH5, ZNF124, ABL1*, and *HNF4A* showing the most significant toxicity (Fig. 2e). We did not observe a preferential dropout or enrichment in commonly amplified vs. commonly deleted genes (Supplementary Fig. S3b), but both OGs and TSGs were depleted more strongly than other genes (Supplementary Fig. S3c). Interestingly, overexpressing OGs in a cancer cell background led to an even stronger viability defect than TSGs, in line with oncogene-induced senescence as a major driver of gene toxicity[42,43]. However, the genes that dropped out the strongest on average were pro-inflammatory genes, indicating that their overexpression is detrimental to cancer cell growth even when cultured in vitro without immune cells (Supplementary Fig. S3d).

Using these screen results, we categorized genes whose overexpression was associated with at least a 30% decrease in growth ($P < 10^{-5}$) as potentially toxic. Among the 12,753 genes tested, 343 genes met these criteria across cancer types and 442 within at least one cancer type (Fig. 2e, f and Supplementary Data 1, 5). The majority of the pan-cancer toxic genes (195) were also found in the tissue-specific analysis (Fig. 2f). To focus on the most confident hits, we considered only genes identified across the pan-cancer analysis (Supplementary Fig. S3e, f).

**Integration of pan-cancer compensation and toxicity analyses identifies ARGOS genes.** Compensated and hyperactivated genes are spread along the genome and are not enriched for frequently amplified genes. The same is true for toxic genes, i.e., genes that drop out in the ORF screens (Fig. 3a). More generally, the distribution of genes of all classes along the genome followed the overall gene density (Supplementary Fig. S4a) and no other strong co-occurrence patterns were observed. Importantly, however, compensated genes were on average also toxic when overexpressed, and hyperactivated genes promoted cell growth and survival in the ORF screens (Fig. 3b and Supplementary Fig. S4b). This is in contrast to genes identified in previous studies, for which compensation did not imply overexpression toxicity (Supplementary Fig. S4c). Whole Genome Doubled (WGD) samples did not show a major difference in compensation or toxicity. Tissue-level scores were correlated at a similar strength to pan-cancer scores as long as there were enough samples available (Supplementary Fig. S4d-h).

Intersecting the sets of pan-cancer compensated and toxic genes yields nine high-confidence ARGOS genes, six of which are also frequently amplified: *RBM12, RBM14, SNRPA, ZBTB14, POU2F1*, and *CDKN1A* (Fig. 3a, c and Supplementary Data 1). As gene dosage compensation has previously been shown to primarily occur at the level of protein complexes[5,9,12], we investigated their functional

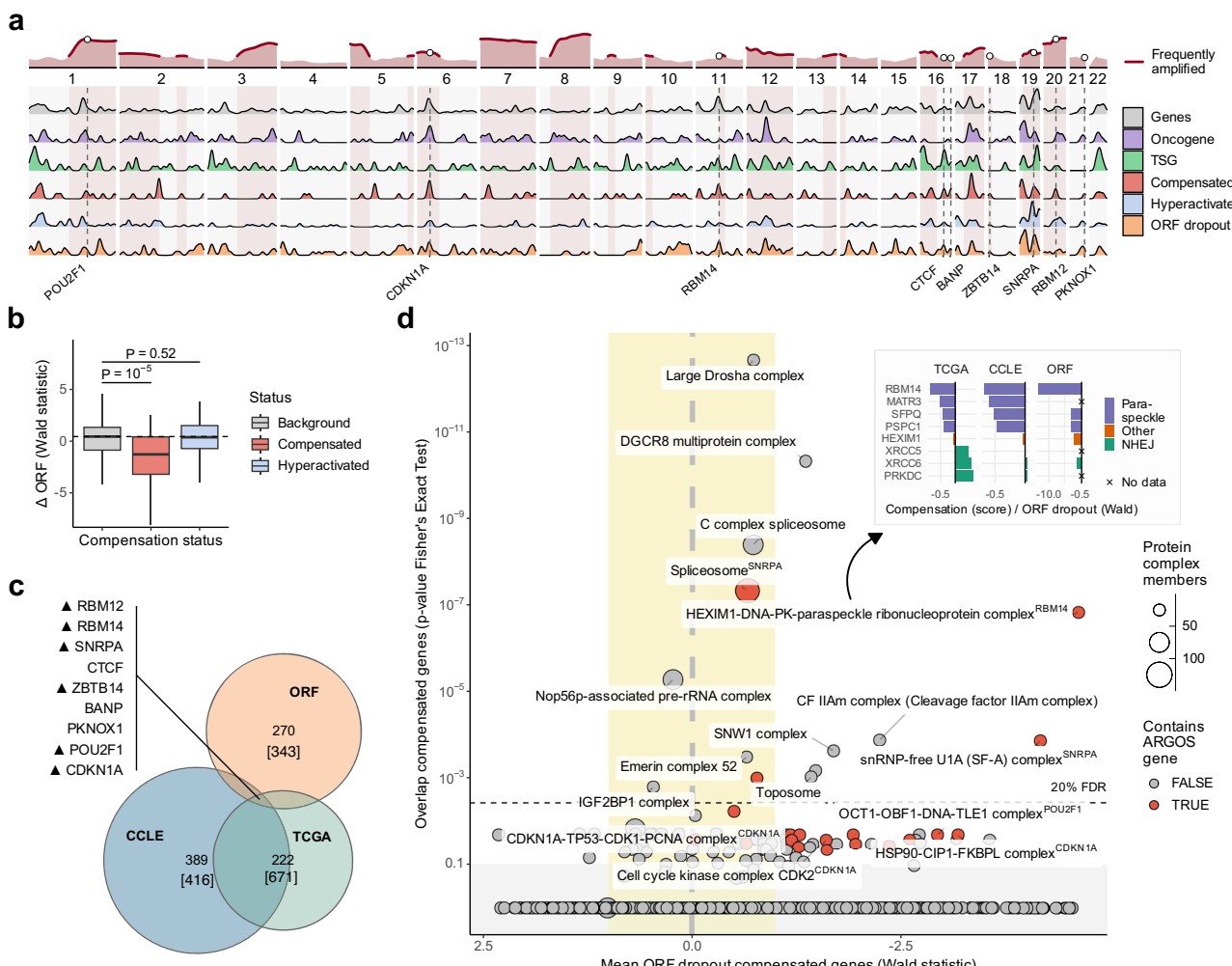

**Fig. 3 | Overlap of compensation and toxicity identifies ARGOS genes. a** All gene classes are spread along the genome and densities using a Gaussian estimator with a 1 Mb bandwidth show clustering only in gene-dense regions (cf. Supplementary Fig. S4a). **b** Compensated genes (110) are dropping out more strongly in the ORF screen than hyperactivated (451) and non-regulated genes. Boxes show median ± quartile, whiskers 1.5x inter-quartile range, and *P*-values from a two-sided *t* test. **c** The overlap of compensated and toxic genes identifies eight genes, five of which are amplified in over 15% of TCGA tumors (upward pointing triangles). Numbers include only genes present in both CCLE and TCGA,

numbers in brackets include all genes present in data. **d** Prioritization of protein complexes with ARGOS genes (Fisher's Exact test on CORUM complexes, *y*-axis) and their toxicity (Wald statistic of a linear regression model for ORF dropout, *x*-axis) identifies the HDP-RNP and U1A spliceosomal complexes. Paraspeckle genes in the HDP-RNP complex are both compensated and toxic, while nonhomologous end-joining (NHEJ) genes show no effect or the opposite trend. Complexes without evidence of compensation and toxicity are shaded gray and yellow, respectively.

impact based on complex membership: Prioritizing both the number of compensated genes within a complex and their degree of toxicity (Fig. 3d), the top 'hits' of this analysis are *RBM14* (part of the HEXIM1-DNA-PK-paraspeckle ribonucleoprotein complex, HDP-RNP; *FDR = 0.00048*) and *SNRPA* (involved in U1A splicing; *FDR = 0.028*). By contrast, general splicing and mi/rRNA processing complexes showed compensation but no toxicity. In addition, *CDKN1A* (p21), a well-characterized TSG and a potent cell cycle inhibitor, showed strong compensation and toxicity based on our analysis but was not significantly enriched in any specific complex (Fig. 3d). To consider both a well-known and a rarely studied gene, we chose *CDKN1A* and *RBM14* for a more detailed investigation. *CDKN1A* is part of the frequently amplified *p* arm of chromosome 6 (Fig. 3a), and its direct role in cell cycle arrest makes it an expected ARGOS gene that can serve as a control for our identification and validation strategies. *RBM14* is located at the edge of a focal amplification next to *CCND1* on chromosome 11, and the nature of its toxicity is unknown. However, its complex members suggest that it may be implicated in the cellular

DNA damage response and in the activation of innate immune signaling (Fig. 3d).

**The well-known cell cycle inhibitor *CDKN1A* as an ARGOS gene**
Amplification of *CDKN1A* occurs at the arm-level in chromosome 6p. Copy number gains of known oncogenes that may drive this arm-level gain[44], such as *CCND3*, *POU5F1*, and *PIM1* reside in its close vicinity (Fig. 4a), suggesting a reason for the common amplification of this tumor suppressor gene. Our analysis identified it as a compensated gene in TCGA and CCLE (Fig. 4b), and our ORF screen data pointed to overexpression of *CDKN1A* resulting in a toxic effect (Fig. 4b, c), in agreement with its important role in cell cycle arrest. To functionally validate the toxic effect of *CDKN1A* overexpression, we selected cancer cell lines from two tumor types (lung and breast): one with normal copy number and average mRNA expression ("non-compensated") and one with lower mRNA/protein expression levels than expected by its amplification ("compensated") (Fig. 4d and Supplementary Fig. S5a). We transduced SK-LU-1 lung

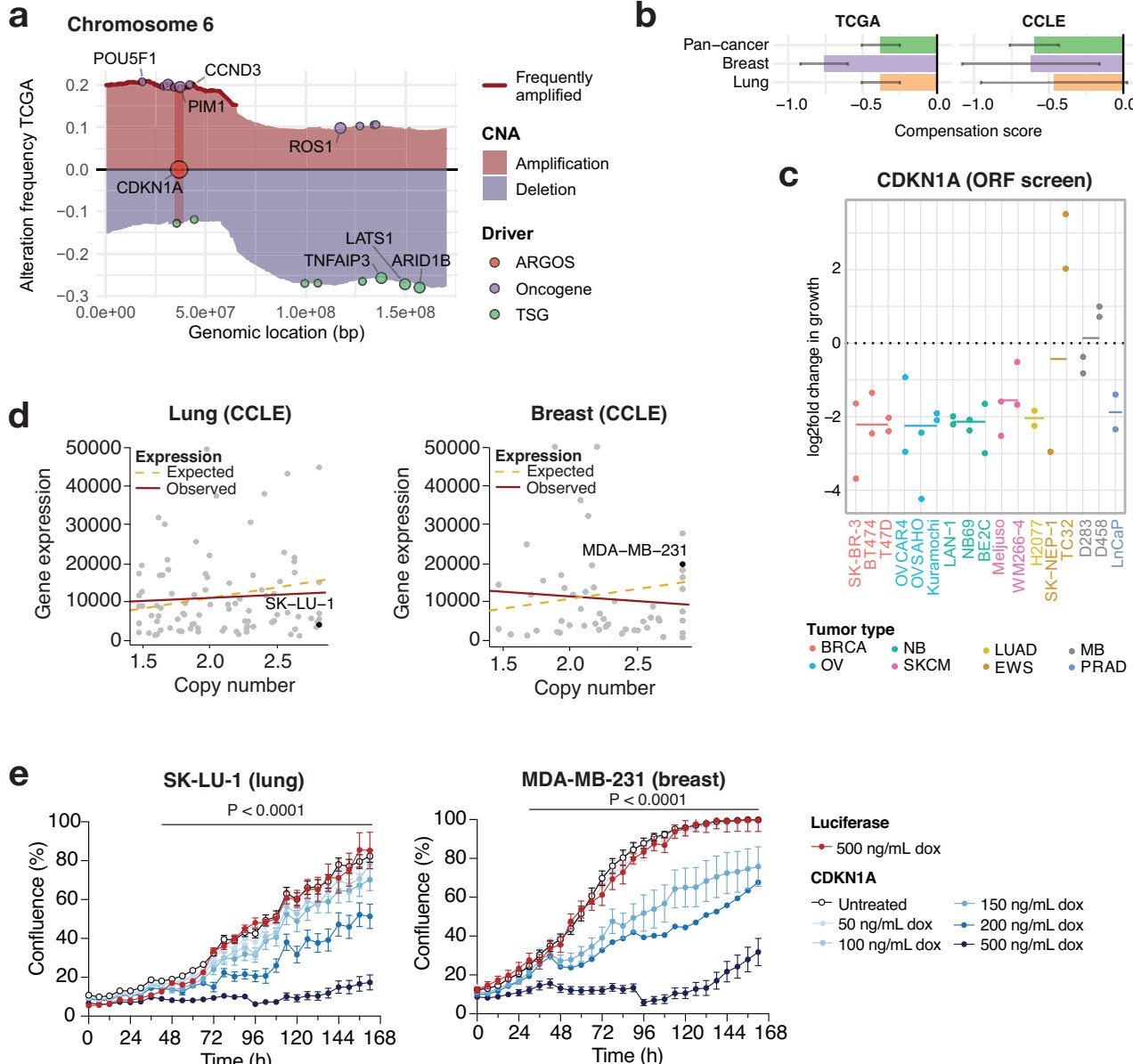

**Fig. 4 | CDKN1A as an ARGOS gene. a** The map of copy number alterations in chromosome 6 shows a p-arm amplification, where *CDKN1A* is located among other known oncogenes. Amplifications (red) and deletions (blue) are shown for this genomic region. The genomic location of OGs and TSGs are shown at the level of amplification- and deletion frequencies for easier visibility, reflecting their respective selection pressure. **b** Compensation scores for *CDKN1A* in our pan-cancer analysis (green) as well as in breast (purple) and lung (orange) lineages chosen for downstream functional validation. Bars represent the mean, error bars the standard deviation of the posterior. **c** Depletion scores for *CDKN1A*

ORFs across 17 independent screens, including breast (BRCA), ovarian (OV), neuroblastoma (NB), skin cutaneous melanoma (SKCM), lung adenocarcinoma (LUAD), Ewing sarcoma (EWS), medulloblastoma (MB) and prostate adeno-carcinoma (PRAD) cell lines. The mean log-fold change in growth is shown as a line for each tumor type. **d** Lung and breast cancer cell lines chosen for functional validation based on their gene expression and DNA copy number profile on CCLE. **e** CDKN1A overexpression leads to a growth inhibition phenotype upon varying levels of overexpression. The mean and S.D. of three replicates are shown. Data was analyzed using two-way ANOVA.

adenocarcinoma and MDA-MB-231 breast cancer cells with doxycy-cline (dox)-inducible and V5-tagged constructs encoding either *CDKN1A* or a luciferase control. By treating cells with a range of dox concentrations (50–500 ng/mL), we were able to detect increases in *CDKN1A* transcript and protein levels (up to 3-fold) for both cell lines in a dose-dependent manner (Supplementary Fig. S5b, c). A luciferase reporter assay confirmed luciferase expression in each cell line model following dox-mediated induction (Supplementary Fig. S5d). To validate the deleterious impact of *CDKN1A* overexpression on cell proliferation, we performed a live-cell imaging assay where we monitored cell confluency in *CDKN1A*- or luciferase-overexpressing cells over time. Indeed, increased levels of gene expression led to a

decrease in cell proliferation, with > 60% growth inhibition observed for SK-LU-1[CDKN1A] and MDA-MB-231[CDKN1A] cells treated with 500 ng/mL dox (Fig. 4e), independent of amplification status. These results are in line with the well-established tumor-suppressive role of CDKN1A[45]. Taken together, these findings serve as proof of concept that we can generate cellular models of gene overexpression to validate ARGOS candidates identified by our compensation and toxicity analyses.

**RBM14 overexpression reduces proliferation of human lung and breast cancer cell lines and causes cell death by apoptosis**
RBM14 belongs to the family of RNA binding proteins, which interact with RNA transcripts to regulate their splicing, cytosolic transport, and

translation[46,47]. RBM14 is known to play a role in alternative splicing regulation of DNA repair via the canonical non-homologous end joining (c-NHEJ) pathway and is part of the HEXIM1-DNA-PK-paraspeckle components-ribonucleoprotein (HDP-RNP) complex that regulates innate immune response through cGAS-STING signaling[48–50]. Its depletion leads to disrupted genome integrity and the accumulation of DNA damage during mouse embryogenesis[51] and hinders mitotic spindle assembly and chromosome segregation in human U2OS bone osteosarcoma cells[52].

*RBM14* gains occur as a focal event in chromosome 11, likely driven by adjacent copy number amplifications of the oncogene *CCND1* (Fig. 5a). Our analysis showed strong gene compensation across cancer types (Fig. 5b), and ORF screens pointed to RBM14 overexpression also leading to detrimental effects on cell proliferation in multiple cancer types (Fig. 5c). To validate the cellular effects of *RBM14* overexpression in human cell lines, we selected two lung adenocarcinoma (NCI-H838; NCI-H1650) and two breast cancer (ZR-75-1; HCC70) cell lines that either exhibit high copy number and lower-than-expected mRNA/protein expression levels ("compensated") or remain euploid at the *RBM14* locus while expressing average levels of gene transcript and protein (Fig. 5d and Supplementary Fig S3a). To generate cell models of RBM14 overexpression, we first transduced cells with a dox-inducible V5-tagged construct encoding either RBM14 protein (generating NCI-H838$^{RBM14}$, NCI-H1650$^{RBM14}$, ZR-75-1$^{RBM14}$, and HCC70$^{RBM14}$) or a luciferase overexpression control (generating NCI-H838$^{luc}$, NCI-H1650$^{luc}$, ZR-75-1$^{luc}$, and HCC70$^{luc}$). By treating all four RBM14-overexpressing cell lines with increasing concentrations of dox (0 to 500 ng/mL), we were able to detect a dose-dependent induction in *RBM14* gene transcript as well as increased expression of V5-tagged exogenous and total RBM14 protein (Supplementary Fig. S6b, c). We confirmed luciferase activity in control cell lines by measuring the luminescent signal in the presence of substrate following dox treatment (Supplementary Fig. S6d).

We next evaluated the impact of *RBM14* overexpression on cell proliferation using live-cell imaging of the RBM14- and luciferase-overexpressing cell lines treated with increasing concentrations of dox. All RBM14-overexpressing cell lines exhibited a dose-dependent reduction in proliferation, with the strongest effects observed as early as 48 h following 500 ng/mL dox induction. By seven days of treatment, NCI-H838$^{RBM14}$, NCI-H1650$^{RBM14}$, ZR-75-1$^{RBM14}$, and HCC70$^{RBM14}$ cells showed > 50% decrease in confluency ($P < 0.0001$) when compared to their respective luciferase controls, supporting the observation that *RBM14* overexpression has inhibitory effects on cell proliferation in vitro (Fig. 5e). This growth inhibitory phenotype was supported by the observation that RBM14 overexpression interfered with nascent protein synthesis, which we measured by incubating cells with a fluorescently labeled methionine analog. Following 48 h of dox induction, we performed immunofluorescence and quantified the amount of average fluorescent signal detected per cell (Fig. 5f). We found that RBM14 overexpression caused a two- to three-fold decrease in fluorescent signal, and hence nascent protein synthesis, when compared to luciferase controls ($P = 0.0106$ for NCI-H838; $P = 0.0011$ for NCI-H1650; $P < 0.0001$ for ZR-75-1; $P = 0.0112$ for HCC70). We next explored whether the reduction in cell proliferation associated with RBM14 overexpression is a result of cell cycle arrest or cell death. Flow cytometry for BrdU incorporation did not indicate significant changes in cell cycle profile between RBM14- or luciferase- overexpressing NCI-H838, NCI-H1650, ZR-75-1, and HCC70 cells (Supplementary Fig. S7a, b). However, when we performed flow cytometry for Annexin V/PI, 20–30% of cells were positive for Annexin V in NCI-H1650$^{RBM14}$ and HCC70$^{RBM14}$ cells by 72 h post-induction with 500 ng/mL dox ($P < 0.0001$) while apoptotic rates remained < 3% in luciferase controls (Fig. 5g). The apoptosis rates of NCI-H838$^{RBM14}$ and ZR-75-1$^{RBM14}$ were lower, with 5–10% cells being positive for Annexin V, but these were still significantly higher rates of apoptosis in comparison to their luciferase controls ($P = 0.025$ for NCI-

H838; $P = 0.0005$ for ZR-75-1). Altogether, these results show that low-level overexpression of RBM14 negatively impacts cellular proliferation and viability in our lung and breast cancer models, regardless of gene amplification status.

## RBM14 overexpression increases reliance on DNA repair by c-NHEJ

Our analysis of compensated protein complexes identified RBM14 as one of several paraspeckle proteins interacting with the DNA-dependent protein kinase (DNA-PK), and other members of the HDP-RNP complex (cf. Figure 3d). This subnuclear complex mediates DNA damage sensing and repair and regulates cGAS-STING signaling and the innate immune response[50,53,54]. We therefore explored whether DNA damage is a mechanism by which RBM14 overexpression might lead to cell death. We first exposed the lung adenocarcinoma NCI-H838$^{RBM14}$/NCI-H838$^{luc}$ and NCI-H1650$^{RBM14}$/NCI-H1650$^{luc}$ cell line pairs to 2 Gy ionizing radiation (IR), and by immunofluorescence measured levels of nuclear γ-H2AX foci as a biomarker of DNA damage. For both cell lines, we were able to detect an initial increase in the number of γ-H2AX foci 15 min after IR, followed by a gradual decrease over time and a return to baseline levels by 360 min after exposure (Fig. 6a). Interestingly, NCI-H838$^{RBM14}$ and NCI-H1650$^{RBM14}$ cells showed a 30–50% reduction of γ-H2AX foci when compared to their respective luciferase controls, with the strongest effect at 60 min post-IR ($P < 0.0001$). We observed a similar decrease in ZR-75-1$^{RBM14}$/ZR-75-1$^{luc}$ and HCC70$^{RBM14}$/HCC70$^{luc}$ pairs of breast cancer cell lines, albeit with a lower effect size (Supplementary Fig. S8a).

Next, we hypothesized that the difference we observed in γ-H2AX phosphorylation dynamics could reflect an increased reliance on RBM14-overexpressing cells on c-NHEJ over homologous recombination (HR) for resolving DNA damage. To assess this hypothesis, we first irradiated cells with 2 Gy and quantified levels of 53BP1 (a marker of the faster c-NHEJ repair) and RAD51 (a marker of the slower HR repair) foci after 60 min and 120 min, respectively. Here, we observed increases in 53BP1 foci following RBM14 overexpression in three out of four cell line pairs, with the largest effect sizes being present in NCI-H838$^{RBM14}$ ($P = 0.0001$) and ZR-75-1$^{RBM14}$ ($P = 0.0094$) cells (Fig. 6b). Conversely, we found significant decreases in RAD51 foci for RBM14-overexpressing cells compared to their relative luciferase controls, particularly in NCI-H838$^{RBM14}$ ($P < 0.0001$), NCI-H1650$^{RBM14}$ ($P = 0.0044$), and HCC70$^{RBM14}$ ($P = 0.0013$) cells (Fig. 6c).

To confirm that RBM14 expression increases the rate of c-NHEJ-mediated DNA repair, we next quantified changes in protein expression of the catalytic subunit of the DNA-PK complex (DNA-PKcs), as auto-phosphorylation at the Ser2056 (pSer$^{2056}$) residue is known to specifically enable c-NHEJ[55]. After 15 min of DNA damage-inducing IR, we found a mean 27.8-fold (range: 0.9–104.8) increase in levels of pSer$^{2056}$ DNA-PKcs in RBM14-overexpressing cells when compared to their respective luciferase controls (Fig. 6d).

Our third experiment leveraged the EJ7-GFP[56] and DR-GFP[57] reporter systems engineered in U2OS cells to quantify the fraction of cells that repair double-stranded breaks (DSBs) by c-NHEJ (based on CRISPR-Cas9 cutting with '7a' and '7b' RNA guides) and HR (based on cutting with the endonuclease I-SceI), respectively. Following the transduction of U2OS cells with RBM14 or luciferase inducible vectors, we transfected cells with plasmids encoding GFP, empty vector controls for both systems and '7a' and '7b' CRISPR guides (c-NHEJ assay) or I-SceI (HR assay). After 72 h, we detected a significant increase in the fraction of GFP-positive U2OS EJ7-GFP RBM14-overexpressing cells repairing DSBs by c-NHEJ ($P = 0.0280$) and a significant decrease in HR-mediated repair in U2OS DR-GFP cells ($P = 0.0142$) compared to luciferase control. We did not detect changes in GFP expression for cells transfected with empty vector controls (Fig. 6c and Fig. S8b).

As c-NHEJ is an error-prone DNA repair pathway[58], we hypothesized that RBM14 overexpression would lead to increased misrepair

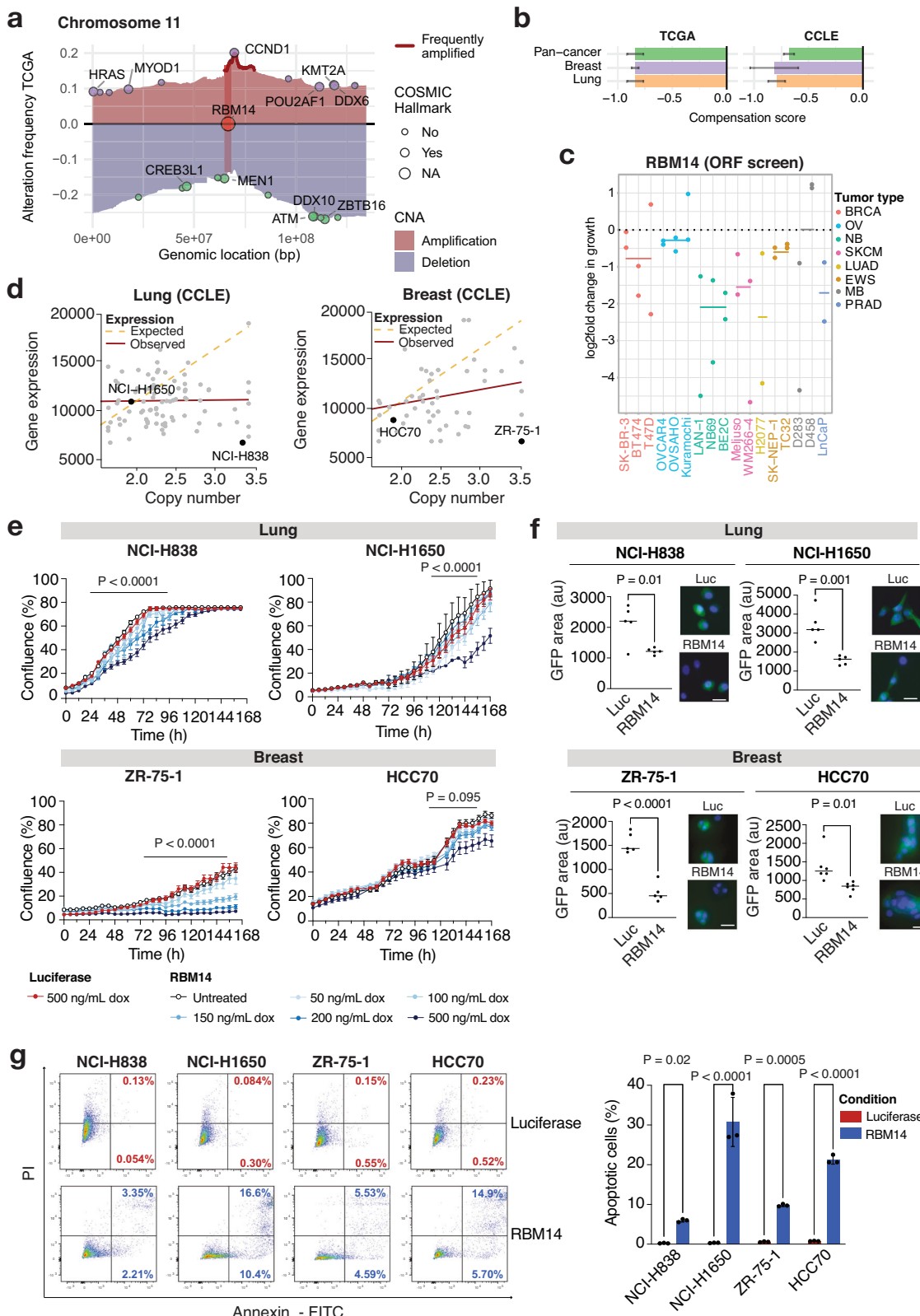

following DNA damage, with resulting genomic instability. To confirm this, we first treated RBM14- and luciferase-overexpressing cells with 2 Gy IR. Next, we used time-lapse microscopy to monitor individual cells during their course of division. Upon induction of DNA damage, we observed a higher percentage of cells with mitotic defects in RBM14-overexpressing cells compared to luciferase controls (*P* = 0.039). The mitotic aberrations detected in RBM14-overexpressing cell lines included the formation of DNA bridges during metaphase, an increase in the number of micronuclei, and incomplete cellular divisions lacking metaphase alignment (Supplementary Fig. S8c, d). These results indicate that RBM14 overexpression leads to a higher rate of aberrant mitotic division following DNA damage and suggest a mechanism by which RBM14 overexpression ultimately leads to cell death in our cell line models.

**Fig. 5 | RBM14 as an ARGOS gene. a** The map of copy number alterations in chromosome 11 shows a focal amplification, where *RBM14* is located next to *CCND1*. Amplifications (red) and deletions (blue) are shown for this genomic region. The genomic location of OGs and TSGs are shown at the level of amplification- and deletion frequencies for easier visibility, reflecting their respective selection pressure. **b** Compensation or ORF dropout scores for *RBM14* in our pan-cancer analysis (green) as well as in breast (purple) and lung (orange) lineages chosen for downstream functional validation. Bars represent the mean, error bars the standard deviation of the posterior. **c** Depletion scores for RBM14 ORFs across 17 independent screens, including breast (BRCA), ovarian (OV), neuroblastoma (NB), skin cutaneous melanoma (SKCM), lung adenocarcinoma (LUAD), Ewing sarcoma (EWS), medulloblastoma (MB) and prostate adenocarcinoma (PRAD) cell lines. The mean log-fold change in growth is shown as a line for each tumor type. **d** Lung and breast cancer cell lines chosen for functional validation based on their gene expression and DNA copy number profile on CCLE. **e** RBM14 overexpression leads to a growth inhibition phenotype. The mean and S.D. of three replicates are shown. Data was analyzed using two-way ANOVA. **f** RBM14 overexpressing lung and breast cells show a decrease in nascent protein synthesis. Fluorescence (GFP; green) is quantified relative to cell number (DAPI; blue). Luciferase overexpressing cells were used as a control. Scale bar: 100 μM. Mean +/− SD is shown from two biological replicates (five images per cell line; six for HCC70) and analyzed by unpaired two-tailed *t* test. **g** The fraction of apoptotic cells following RBM14 overexpression was determined by flow cytometry-based quantification of annexin V and PI-positive cells. The mean and S.D. of three replicates is shown. Data was analyzed by unpaired two-tailed *t* test. Source data for Fig. 5c, f, and g are provided as Source Data files.

Finally, we explored whether the increased dependency of RBM14-overexpressing cells on c-NHEJ-mediated DNA repair exposed new vulnerabilities in these cells. For this purpose, we tested the sensitivity of our cell line models to the DNA-PK inhibitor NU7441 (thereby inhibiting c-NHEJ repair) and the RAD51 inhibitor B02 (inhibiting HR-mediated repair) after inducing DNA damage by 2 Gy IR. All four RBM14-overexpressing cell lines showed increased sensitivity to DNA-PK inhibition when compared to luciferase controls after 3 days of treatment, albeit with a mild effect size (Fig. 6d). By contrast, RBM14 overexpression did not alter the sensitivity to RAD51 inhibition in three out of the four cell lines (Supplementary Fig. S8e). Altogether, these results support an increased dependency of RBM14-overexpressing cells on c-NHEJ over homologous recombination (HR) for resolving DNA damage, which underlies RBM14 toxicity and may suggest therapeutic sensitivities associated with its expression.

## RBM14 overexpression modulates STING-STAT3 signaling and enhances NK cell-mediated tumor recognition

As part of the HDP-RNP complex, RBM14 also plays an important role in recognizing the accumulation of cytoplasmic DNA and is required for the production of type I interferons following STING pathway activation[50,59,60]. We therefore hypothesized that, in the presence of DNA damage, RBM14 overexpression would also lead to increased recognition of cytosolic DNA strands and activation of STING signaling. To assess this, we exposed NCI-H838[RBM14]/NCI-H838[luc] and NCI-H1650[RBM14]/NCI-H1650[luc] overexpressing cells to 2 Gy IR and measured by immunofluorescence of the perinuclear localization of STING as a marker of STING pathway activation[61]. Indeed, in both cell lines, RBM14 overexpression led to a significant increase in perinuclear localization of STING (quantified as the relative STING area outside the cellular nucleus; $P = 0.045$ for NCI-H838[RBM14] and $P < 0.0001$ for NCI-H1650[RBM14]), whereas no effect was observed in luciferase-overexpressing control cells (Fig. 7a, b).

We next explored whether RBM14 overexpression modulates downstream STING signaling in our cell line models. We have recently shown how inflammatory responses in chromosomally unstable breast cancers are accompanied by increased STAT3 signaling, which is in itself dependent on STING pathway activation[62] and has been linked to immunosuppressive phenotypes[63]. To determine whether RBM14 affects the STING/STAT3 signaling axis, we evaluated the effect of gene overexpression on cellular response to STAT3 inhibition. To this end, we induced DNA damage in our set of RBM14- vs. luciferase-overexpressing lung and breast cancer cell lines and treated with the STAT3 inhibitors C188-9 and HJC052 for 72 h. We found that RBM14-overexpressing cells treated with 1 μM of the STAT3 inhibitors C188-9 and HJC052 were less affected by these inhibitors compared to the luciferase controls (Fig. 7c). However, it should be noted that RBM14-overexpressing cells had a lower starting viability when compared to luciferase control cells (Supplementary Fig. S9a, b). These results suggest a potential role for RBM14 in impairing STAT3 signaling, leading to lower viability and decreased sensitivity to STAT3 inhibitors while contributing to an inflammatory phenotype in our cell line models.

To determine whether RBM14-dependent activation of STING can also elicit non-cell-autonomous immune responses, we finally performed co-culture experiments with our cancer cell line pairs and natural killer (NK) cells. To this end, we irradiated RBM14- and luciferase-overexpressing cancer cells with 2 Gy and added NK-92 cells to the culture at a 10:1 ratio of 2 h following irradiation. After 48 h of co-culture, we removed the NK-cells, fixed and stained the cancer cells with crystal violet, and quantified the area of colony formation as a measurement of cancer cell viability (Fig. 7d). A reduction in colony area was observed for all co-cultured cancer cell lines compared to control conditions #1 (only cancer cells cultured in regular cancer cell media) and #2 (only cancer cells cultured in NK-92 cell media). Relative to luciferase controls, we observed decreases in colony area for both lung NCI-H838[RBM14] ($P = 0.5308$) and NCI-H1650[RBM14] ($P = 0.5138$) cells, and a significant reduction in colony area (> 60%) for HCC70[RBM14] cells ($P = 0.0192$).

In sum, our results show how RBM14 overexpression promotes STING activation in the context of DNA damage, and how this effect impairs immunosuppressive STAT3 signaling and may enhance NK cell-mediated tumor recognition and killing.

## RBM14 amplification status is correlated with survival in a clinical CRC cohort treated with IR

To determine whether RBM14 amplification and overexpression can be clinically actionable, we identified a clinical study cohort of 1063 colorectal cancer (CRC) patients, all treated with IR as standard of care[64]. Of those, 110 had a *CCND1* amplification, 95 a *RBM14* co-amplification and 952 had neither. We confirmed a stronger dosage-sensitive scaling for *CCND1* (Fig. 8a) compared to *RBM14* (Fig. 8b), indicating that also in this cohort *RBM14* expression is partially compensated. We observed that *CCND1* amplifications themselves were negatively correlated with overall patient survival, while the combined amplification with *RBM14* was positively correlated ($P = 0.048$, Cox Proportional Hazards model; Fig. 8c). These results suggest that *RBM14* amplifications indeed drive a cancer vulnerability, clinically exploitable by IR.

## Discussion

Current targeted therapies in cancer are designed against a handful of oncogenic proteins that drive disease progression. In an effort to expand the universe of potentially exploitable anti-cancer targets, recent genomic efforts have started to uncover therapeutic vulnerabilities arising from copy number losses not only affecting driver but also passenger genes in cancer[13,14]. However, it is much less known how much genes collaterally affected by amplifications can drive similar sensitivities[20,24]. In this study, we sought to define and characterize the class of 'Amplification-Related Gain Of Sensitivity' (ARGOS) genes, causing a proliferation defect in cancer cells when overexpressed, often due to their close proximity to focal oncogenic driver amplifications.

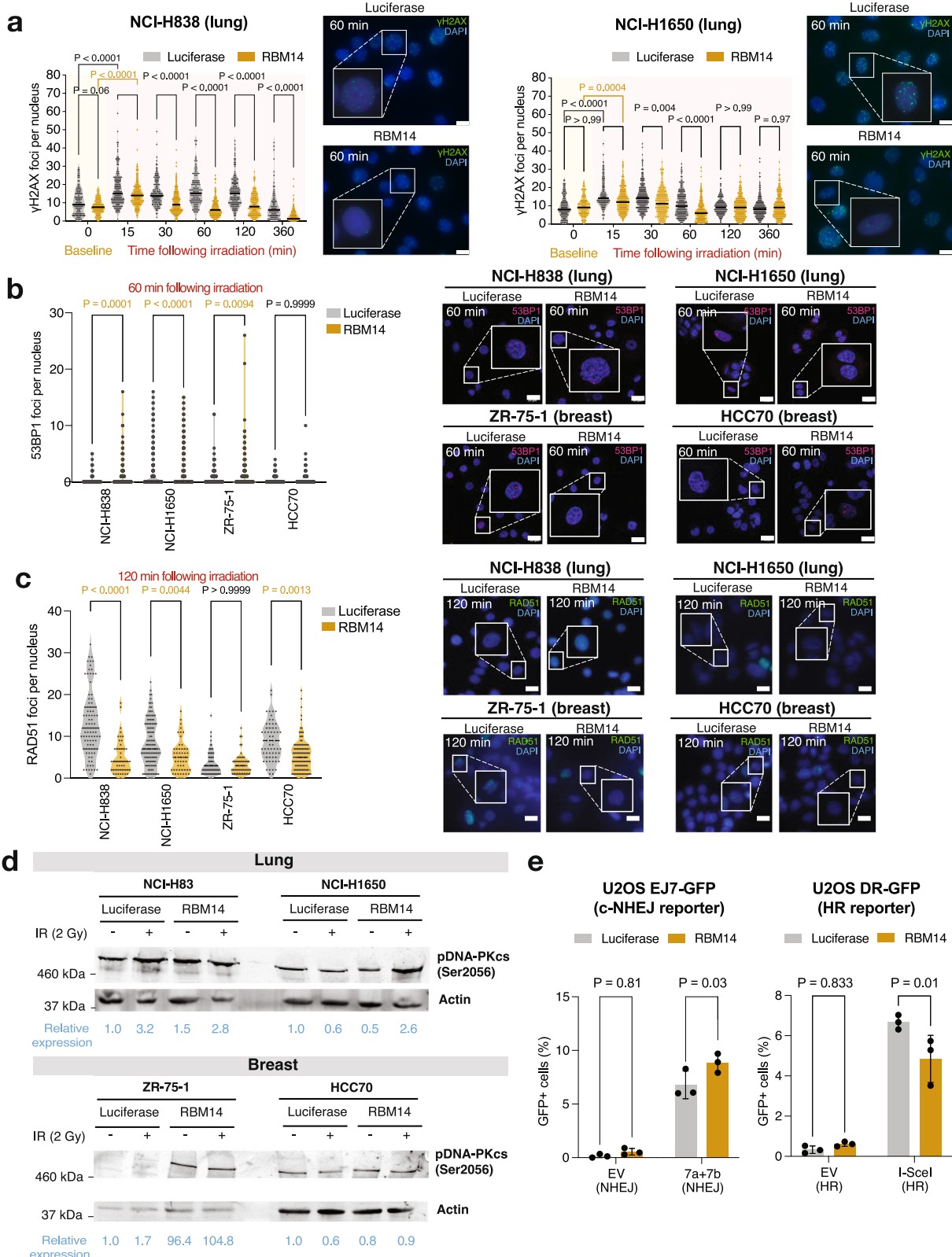

Gene expression upon amplification is generally considered dosage-sensitive[4], but previous studies have shown extensive compensation of protein expression[5,9,65,66]. To measure gene expression compensation, studies have so far quantified the degree of over-expression with genomic amplifications[9,11,67] or changes in their correlation[67]. Here, we instead quantified the evidence for a deviation of gene expression from the expected slope of dosage-sensitive

scaling. We preferred using this approach, so that genes that scale with copy number but are noisy in their expression levels would not be falsely identified as compensated. Similarly, we preferred using mRNA over protein expression, as the available data are more complete and RNA levels are more strongly correlated with DNA levels than proteins.

Our ARGOS genes represent a class of sensitivities that would go unnoticed in more traditional efforts to identify dependency genes,

**Fig. 6 | RBM14 overexpression leads to differential DNA damage response.**
**a** The number of yH2AX foci was quantified by immunofluorescence in lung NCI-H838[RBM14/luc] and NCI-H1650[RBM14/luc] cells following 2 Gy ionizing radiation (IR). Data from three biological replicates analyzed by two-way ANOVA. The horizontal bar in each violin plot indicates the mean. Representative images are shown for each condition at 60 min following IR. Scale bar: 10 μM. **b, c** Quantification of (**b**) 53BP1 and (**c**) RAD51 foci in RBM14- and luciferase-overexpressing cells at 60 min and 120 min post 2 Gy irradiation, respectively. Data was analyzed by two-way ANOVA. Representative images are shown for each condition. Scale bar: 20 μM. **d** Levels of DNA-PKcs (pSer2056) were detected and quantified by immunoblotting in cells at 15 min following 2 Gy IR. Representative immunoblot from two biological replicates. **e** RBM14 overexpressing U2OS cells exhibit higher rates of c-NHEJ-mediated repair. The fraction of U2OS[RBM14/luc] cells that repair double-strand breaks by c-NHEJ (EJ7-GFP; left) or HR (DR-GFP; right) was determined by quantifying GFP-positive cells in a flow cytometry-based reporter assay. GFP-positive cells were normalized to GFP transfection controls. Empty vector (EV) controls were included for comparison. Data from three biological replicates are presented as mean +/− SD and analyzed with a two-sided *t* test. Source data for Fig. 6a–d are provided as Source Data files.

such as RNA interference or CRISPR-Cas9 mediated knock-out screens. They are complementary to previously identified genetic dependencies correlated with copy number gains in cell lines[12,23,24], as evidenced by the fact that previous studies did not identify the *CCND1-RBM14* co-amplification[24]. Hence, this joint computational and experimental approach enables the rational identification and prioritization of compensated genes with strong biological evidence for their toxicity. We provide the identified compensated, toxic, and ARGOS gene lists as a resource to the community for follow-up in individual as well as across cancer types (Supplementary Data 1–5).

We selected *CDKN1A* and *RBM14* as proof-of-concept candidates to identify putative mechanisms of toxicity arising from genomic amplification. For this, we engineered cell line models where protein overexpression could be induced at levels equivalent to those achieved by copy number gains present in TCGA and CCLE samples. Whereas a strong toxicity and compensation profile could be expected for the tumor suppressor and cell cycle regulator CDKN1A, the effects of RBM14 overexpression on cancer cell growth are novel. Previous studies identified *RBM14* as an essential gene during mouse embryogenesis[51], with its genetic depletion leading to increased genomic instability and cell death by apoptosis. In addition, RBM14 has been implicated in DNA repair by c-NHEJ[49,68], and its knock-out sensitizes glioblastoma cells to radiotherapy[69]. Here, we find that overexpression of RBM14 at similar levels to commonly observed gains can already cause a detrimental effect: (1) gene overexpression leads to preferential DNA damage repair by c-NHEJ, an error-prone process that increases the rate of aberrant cell divisions; (2) upregulation of RBM14 and DNA-PK as members of the HDP-RNP complex induces cGAS-STING pathway activation.

In accordance with protein stoichiometry imbalances driving a common phenotype[70], the toxicity effects in our cell line models were observed regardless of RBM14 gene amplification status. Interestingly, the proliferation defect via RBM14-induced STING activation and STAT3 modulation is in line with the strong enrichment for immune response pathways observed in genes that drop out in the analyzed ORF screens, supporting its putative role as a cell-intrinsic driver of toxicity. RBM14 overexpression also led to increased NK cell recognition and killing in our co-culture experiments, suggesting an additional, non-cell-autonomous mechanism for RBM14 toxicity. However, we do not discard the possibility that other processes beyond DNA damage response and innate immune signaling contribute to RBM14's mechanism of toxicity. This is because both our ORF screen toxicity analysis and initial validation of lung and breast cancer cell lines showed gene overexpression to strongly decrease cellular proliferation and protein translation rates in the absence of IR-mediated DNA damage. Identification of all biological processes associated with RBM14 overexpression, as well as their relative contribution, will require further investigation in additional cancer cell line models as well as in vivo experiments.

Finally, our compensation and toxicity analyses identified 7 additional ARGOS genes that remain to be validated. However, this number may be an underestimation of the full universe of targets, as (1) the ORF screens we analyzed were performed with lentiviral libraries that only span about half of the protein-coding genome (12,753 genes) and were

conducted in a limited number of tumor types; (2) some genes may be compensated only at the protein level and not at the mRNA level; and (3) we were very stringent with our definitions of ARGOS genes in this proof-of-concept study.

Overall, our study establishes a compendium of genes compensated across human cancers and in tissue-specific patterns, with functional evidence for toxicity effects associated with gene overexpression in the context of this disease. We predict that additional candidates will be discovered as data from other functional genomic approaches (such as CRISPR activation screens) becomes more widely available. As ARGOS genes can constitute cellular liabilities, their discovery could ultimately lead to the emergence of therapeutic opportunities in tumors harboring ARGOS gene amplification. We have provided an example of this with *RBM14* amplifications in an IR-treated CRC cohort.

## Methods

### Gene compensation
The gene compensation analysis was performed either across tissues (pan-cancer) or for individual cancer types (tissue-specific), both for TCGA and CCLE data. Genes were considered frequently amplified or deleted if they were identified by GISTIC in 15% or more of TCGA samples. Both CCLE and TCGA RNA-seq library size factors were calculated from raw counts using *DESeq2* (1.31.3)[71]. Copy number counts were transformed from $\log_2$ into linear space and parameterized in euploid equivalents $c$ and euploid deviation $d$. Therein, $c$ represents the slope, i.e. the part of gene expression that is scaling with DNA copy number (0 if there are no DNA copies present, 1 if it is equal to the sample mean). $d$ represents the deviation from this slope, with a value of 0 for the sample mean, −1 if there are no DNA copies present, or 1 if there are twice the base copies. Hence, $c$ is always one integer unit higher than $d$. The expression scaling term $c$ is fitted per tissue $t$, allowing for different basal expression levels per tissue. This leaves us, for the CCLE, with the regression formula for each gene across samples:

$$\mu = \Sigma_t \quad \beta_1{}^t c + \beta_2 \, d + \varepsilon \qquad (1)$$

For the TCGA, we multiplied the cancer component of the gene expression by its purity term $p$, and add an additional term to represent the gene expression contribution of the non-cancer compartment:

$$\mu = \Sigma_t \quad \beta_1{}^t c \, p + \beta_2 \, d \, p + \Sigma_t \quad \beta_3{}^t (1 - p) + \varepsilon \qquad (2)$$

Both formulas are equivalent for a purity value of 1, as in this case $\beta_3$ becomes zero and in the first two $p$ terms can be dropped. Note that for TCGA $c$ and $d$ are DNA copies for cancer cells, hence $c\,p$ and $d\,p$ represent the observed DNA copy number in the mixture. Each $\mu$ represents the mean gene counts observed for a given gene in RNA-seq data divided by library size. For each gene, we then fit a Negative

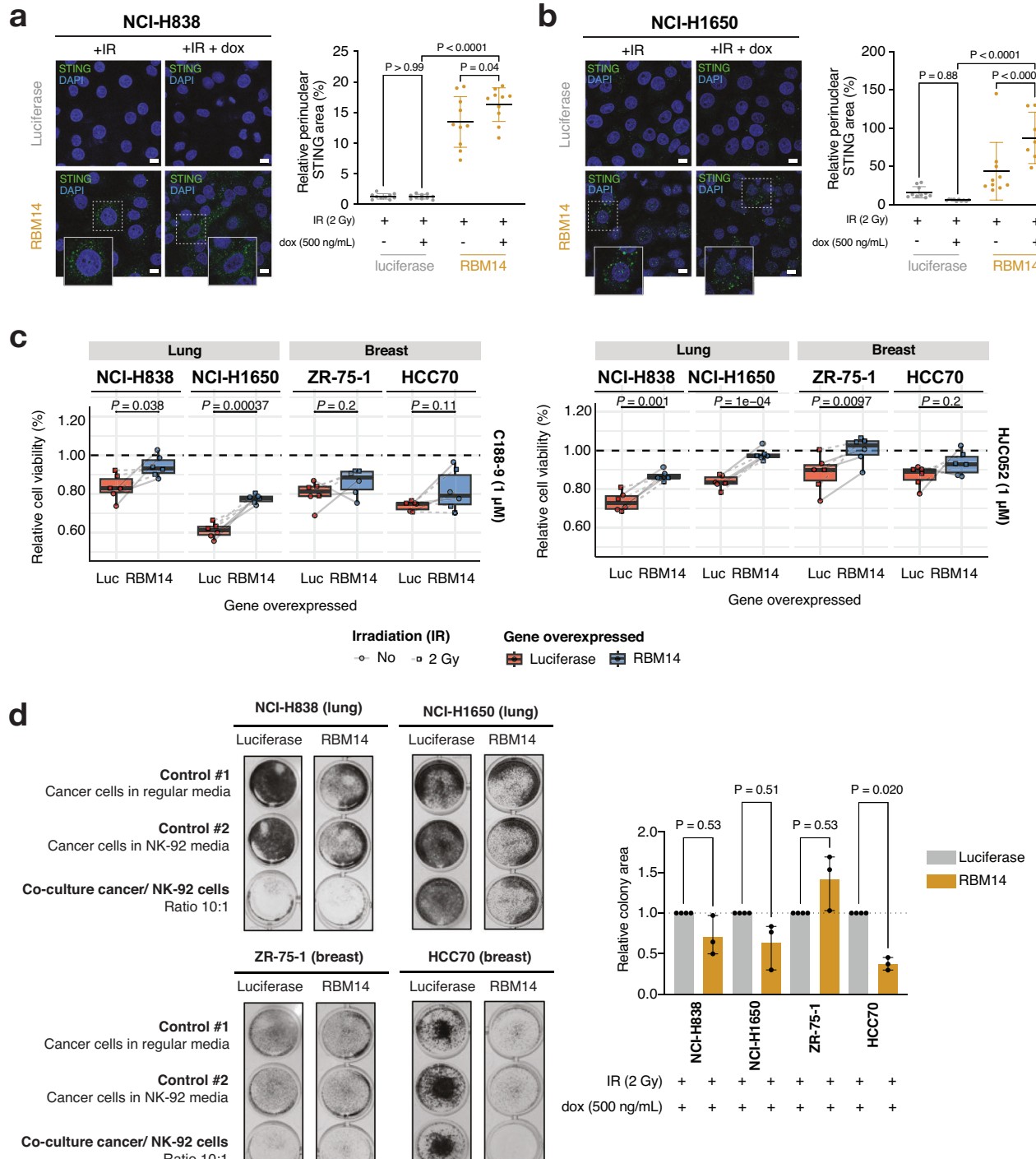

**Fig. 7 | RBM14 overexpression induces a STING-mediated innate immune response. a, b** Immunofluorescence quantification of perinuclear STING area in RBM14- and luciferase- overexpressing (**a**) NCI-H838 and (**b**) NCI-H1650 cells. STING area (mean +/− SD as indicated) was calculated relative to the total nuclei area in a total of 10 images per condition (from three biological replicates). Data was analyzed with two-way ANOVA. Scale bar: 10 μm. **c** Cell viability of RBM14- (blue) and luciferase- (red) overexpressing lung and breast cancer cell lines after 72 h of treatment with 1 μM C188-9 (left) and HJC052 (right) STAT3 inhibitors. Viability is shown relative to DMSO (0.1 μM for HCC70) treatment condition for

each cell line (in three replicates for each condition) in the absence (circled symbol; solid line) or presence (squared symbols; dashed line) of 2 Gy DNA damage-inducing irradiation (IR). Boxes show median ± quartile, whiskers 1.5x inter-quartile range, and *P*-values from a two-sided *t* test. **d** Relative colony area (mean +/− SD) of RBM14- and luciferase- overexpressing lung and breast cancer cell lines after co-culture with NK-92 cells at a 10:1 ratio for 48 h following 2 Gy irradiation. Data was analyzed with one-way ANOVA three biological replicates. Source data for Fig. 7a−d are provided as Source Data files.

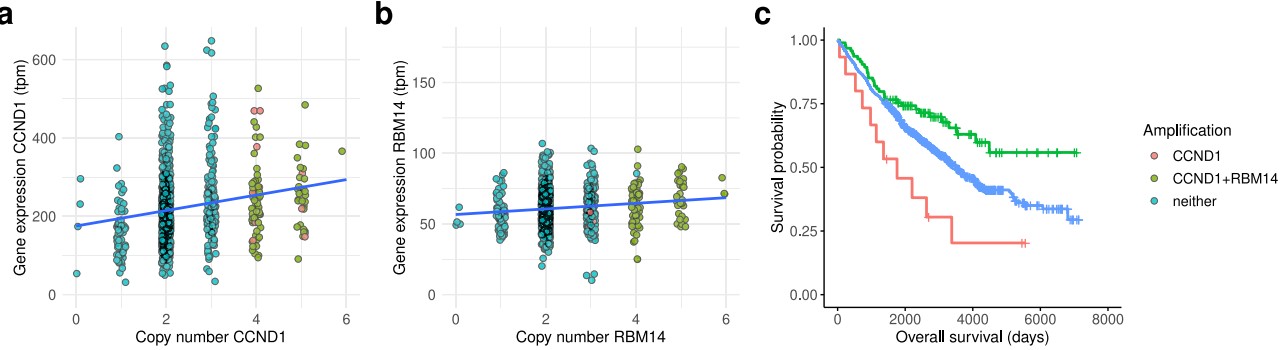

**Fig. 8 | Impact of CCND1/RBM14 amplifications on patient survival in a CRC cohort. a** Copy number and gene expression of *CCND1* and (**b**) *RBM14*. **c** Overall patient survival by co-amplification of *CCDN1* and *RBM14* vs. only *CCND1*. Cox Proportional Hazards model adjusting for sex and age $P = 0.048$.

Binomial regression:

$$r \sim NB(s\,m\,\mu, \sigma) \qquad (3)$$

Here, for the total number of observed reads $r$ and in addition to a theoretical gene mean $\mu$ we need to take into account the RNA-seq library size $s$, as well as the mean expression of a gene $m$ (so we obtain comparable parameters independent of gene expression level and library size). We fit this regression using the R package *brms* (version 2.20.1)[72] using a tissue-specific scaling term ($\beta_1$), a common compensation term ($\beta_2$), in the case of TCGA data a tissue-specific purity term ($\beta_3$), and a linear link function for $\beta$. $\sigma$ represents the shape parameter. We used the default prior for $\sigma$ and set the following priors on $s\,m\,\mu\,\beta_i$:

$$\beta_{1,3} \sim logN(0, 1) \qquad (4)$$

$$\beta_2 \sim N(0, 0.5)\ \text{for CCLE} \qquad (5)$$

$$\beta_2 \sim N(0, 0.2)\ \text{for TCGA} \qquad (6)$$

We included from the CCLE/TCGA all samples where a gene has no copy number changes (15% tolerance) and at least 3 or 5 samples with an amplification, respectively (1 copy number with 15% tolerance). We compared our proposed model with a model lacking a purity term (Supplementary Fig. S2a, left) and a model with a common purity term across tissues (Supplementary Fig. S2a, center), and in both cases found an inflation of compensation in the TCGA over the CCLE, which is why we did not use these models. As proposed, the result in TCGA and CCLE showed the strongest correlation without an obvious inflation effect in the TCGA (Supplementary Fig. S2a, right). We normalized $\beta_2$ by the mean euploid expression ($\underline{\beta}_1^t$) and calculated statistical significance using the z-score between the posterior and the origin of $\beta_2$. To arrive at the final compensation score $s$, we shrank $\beta_2^*$ by its pseudo-$p$-value derived from the z-score:

$$\beta_2^* = \beta_2 \div \underline{\beta}_1^t \qquad (7)$$

$$s = (1 - p) \times \beta_2^* \qquad (8)$$

We rely on this shrunk coefficient as well as the TCGA-CCLE overlap to confidently assign gene compensation. We estimated gene-set level differences using a linear model on the compensation scores of each gene in a set.

**Protein compensation**
We used normalized protein expression data from CCLE proteomics[73] and reverse log$_2$-transformed them back into a linear scale. We then estimated the expected scaling between no expression without DNA copies and observed average expression (1 for normalized data), as well as observed scaling using a linear regression model.

**ORF screen analysis**
The ORF screen ("toxicity") analysis was performed in both a pan-cancer and tissue-specific manner. We repurposed the vehicle treatment arm of 17 independent ORFeome library screens that were conducted to assess drug sensitivity across eight tumor types (Genetic Perturbation Platform, Broad Institute). For each screen, cells were infected with the ORFeome pLX317 barcoded library (16,100 barcoded ORFs overexpressing 12,753 genes), selected with puromycin, and cultured for 14–21 days. We compared the high-throughput sequencing data from early (after puromycin selection) and late (after 14–21 days of cell culture) time points to estimate log$_2$-fold changes and significance in ORF barcode representation using a linear model, where each barcode measurement for a given gene in any screen was considered an independent observation. All screens were included for the pan-cancer analysis, whereas the tissue-specific analysis was limited to screens that share a cancer type. We estimated gene-set level differences using a linear model on the Wald statistic of each gene in a set.

**Cell line information and tissue culture**
NCI-H838 (cat. no. CRL-5844; ATCC; male), NCI-H1650 (cat. no. CRL-5883; ATCC; male), SK-LU-1 (cat. no. HTB-57; ATCC; female), ZR-75-1 (cat. no. CRL-1500; ATCC; female), HCC70 (cat. no. CRL-2315; ATCC; female), and MDA-MB-231 (cat. no. HTB-26; ATCC) cells were initially STR-profiled and subsequently cultured in RPMI 1640 medium (Gibco) supplemented with 10% tetracycline negative fetal bovine serum (FBS; Gemini Bio) and 1% penicillin-streptomycin (Invitrogen). HEK-293T cells (cat. no. CRL-11268; ATCC) were grown in DMEM (Gibco) with 10% FBS. U2OS EJ7-GFP and DR-GFP cells were a gift from Dr. Jeremy Stark (City of Hope) and were cultured in McCoy's 5 A medium (Gibco), supplemented with 10% FBS and 1% penicillin/streptomycin. All cell lines were maintained at 37 °C in 5% $CO_2$ and frequently examined for mycoplasma contamination using the MycoAlert Mycoplasma Detection Kit (Lonza). Cell lines used in experiments tested negative for mycoplasma contamination.

**Plasmid construction**
Open reading frame (ORF) constructs were obtained from the Genetic Perturbation Platform (Broad Institute). Entry pDONR223 vectors for CDKN1A (clone ID ccsbBroadEn_00282), RBM14 (clone ID ccsbBroadEn_02429) and luciferase control (clone ID BRDN0000464768) were cloned into the pLXI-403 dox-inducible destination vector (clone ID BRDN0000464768) by Gateway®-compatible LR clonase II (Invitrogen). This final expression vector contained the coding sequence of each gene followed by an in-frame V5-tag sequence. Following

Reaction products were transformed in One Shot Stabl3 chemically competent *E. coli* (Invitrogen) under 100 μg mL⁻¹ carbenicillin selection (Sigma-Aldrich). Plasmid DNA from single colonies was extracted with the QIAprep Spin Miniprep Kit and Maxi Kits (Qiagen) and confirmed by Sanger sequencing (Genewiz) using the primer 5′-G AC GTG AAG AAT GTG CGA GA-3′.

### Lentiviral transduction

Lentivirus-compatible expression vectors generated from Gateway Cloning were transfected in HEK-293T cells with Lipofectamine 3000 Transfection Reagent (cat. no. L3000001, Invitrogen) along with psPAX2 and pVSV-G viral packaging plasmids (Addgene) following manufacturer's protocol. The virus was harvested 48 hours later with a 0.45-micron syringe filter. For transduction of viral plasmids, cells were seeded at a density of 2 million cells/well in a 12-well plate. Virus (400 μL) was added together with 5 μg/mL of polybrene (Sigma-Aldrich) and cells were centrifuged at 300 × *g* for 2 h at 30 °C. The next day, cells were selected with 1 μg/mL puromycin (Gibco).

### Quantitative reverse transcription PCR (RT-qPCR)

One million cells were treated with 0–500 ng/mL dox for 5 h. Following treatment, RNA extraction was conducted using the RNeasy Mini Kit (cat. no. 74004, Qiagen) with on-column DNase digestion (cat. no. 79254, Qiagen). Next, 2 μg of purified RNA was converted to cDNA with the Maxima H Minus first-strand cDNA synthesis kit (Thermo Fisher), using the random hexamer primer provided per manufacturer's protocol. For RT-qPCR analysis, 500 μg of resulting cDNA was added to a reaction mixture containing 1X Maxima SYBR Green/ROX qPCR Master Mix (cat. no. K0221, Thermo Fisher Scientific) and 0.3 μM forward and reverse primers (IDT) against CDKN1A (forward 5′-GGA AGA CCA TGT GGA CCT GT-3′; reverse 5′-GGA TTA GGG CTT CCT CTT GG-3′), RBM14 (forward 5′-TTT TCG TGG GCA ATG TGT CGG C-3′; reverse 5′-GAT TGC GGC TTT GGC ATC TGC T-3′), and actin (forward 5′-CCG AAA GTT GCC TTT TAT GG-3′; reverse 5′-TCA TCA TCC ATG GTG AGC TG-3′). Samples were run in a StepOne RT-qPCR thermocycler, using the following running protocol: 95 °C for 15 s and 60 °C for 1 min for 40 cycles, with melt curve stage at 95 °C for 15 sec and 60 °C for 1 min. Statistical analysis was performed by the $\Delta\Delta C_T$ method using the StepOne Software v2.1 (Applied Biosystems). For the analysis of STING downstream targets, the following forward and reverse primers were used: IL-6 (forward 5′-CAGGAGCCCAGCTATGAACT-3′; reverse 5′-GAA GGC AGC AGG CAA CAC-3′), IL-8 (forward 5′-TTT TGC CAA GGA GTG CTA AAG A-3′; reverse 5′-AAC CCT CTG CAC CCA GTT TTC-3′), CXCL10 (forward 5′-GAA AGC AGT TAG CAA GGA AAG GT-3′; reverse 5′-GAC ATA TAC TCC ATG TAG GGA AGT GA-3′), and CCL5 (forward 5′-TGC CCA CAT CAA GGA GTA TTT-3′; reverse 5′-CTT TCG GGT GAC AAA GAC G-3′).

### Immunoblotting

Cells were treated with 0–500 ng/mL doxycycline (Takara Bio) 48 h prior to sample collection. Pellets from 2 million cells were collected and lysed with RIPA lysis buffer (20 mM Tris-HCl at pH 7.5, 150 nM NaCl, 1 mM Na₂EDTA, 1 mM EGTA, 1% NP-40, 1% sodium deoxycholate, 2.5 mM sodium pyrophosphate, 1 mM beta-glycerophosphate, 1 mM Na₃VO₄, 1 μg/mL leupeptin), phenylmethanesulfonyl fluoride (PMSF; Cell Signaling Technology) and a protease and phosphatase inhibitor cocktail (Boston BioProducts). Lysates were incubated for 45 min on ice with pulse-vortexing in 15 min intervals and centrifuged at 14,000 × *g* for 10 min in 4 °C. Protein concentrations were quantified by BCA against a bovine serum albumin standard (ThermoFisher). Absorbance was measured by incubating protein samples and BCA standards with ƒcolorimetric reagents (Thermo Scientific). A total of 25 μg for each sample was loaded on a NuPAGE 4–12% Bis-Tris gel with 1X NuPAGE MOPS Running buffer (Life Technologies, Invitrogen) and separated for 1.5 h at 120 V. For protein size comparison, the Cytiva full-range Rainbow™ molecular weight marker (Fisher Scientific) was

used as reference. Following protein separation, a dry transfer was completed at 30 V for 6 min with PVDF stacks in an iBlot 2 transfer instrument (Thermo Fisher). After gel transfer, the membrane was blocked in 5% dry milk in TBS-T for 30 min and incubated overnight at 4 °C with primary antibodies against: rabbit V5 at 1:1,000 (cat. no. 13202S, Cell Signaling Technology), mouse vinculin at 1:1,000 (cat. no. V9131, Millipore Sigma), rabbit CDKN1A at 1:1,000 (cat. no. 2947S, Cell Signaling Technology), and rabbit RBM14 at 1:1,000 (cat. no. Ab70636, Abcam) in blocking solution containing 5% milk and TBS-T. For the detection of phospho-DNA-PKcs, samples were run in a 3–8% Tris-acetate gel with 1X NuPAGE MES Running buffer (Life Technologies, Invitrogen). A wet transfer was done overnight at 30 V, and the membrane was blocked in 5% BSA in TBS-T. Overnight incubation was done with primary antibodies against rabbit DNA-PKcs phospho S2056 (cat. no. ab124918, Abcam) and mouse actin at 1:1000 (cat. no. 4970, Cell Signaling Technology). After incubation with primary antibodies, the membranes were washed in TBS-T and further incubated for 1 h with goat anti-rabbit at 1:3000 (cat. no. 7074S, Cell Signaling TechnologyThermoFisher) and goat anti-mouse (cat. no. 7076S, Cell Signaling TechnologyThermoFisher) at 1:10,000 dilution or anti-rabbit at 1:3000 (IRDye 700CW, LiCor) and anti-mouse (IRDye 800CW, LiCor) antibodies at 1:3000 dilution for DNA-PKcs phospho S2056. The signal was detected by chemiluminescence with SuperSignal West Pico and Femto kits (Thermo Scientific) in ImageQuant LAS 4000 imager (GE Healthcare Life Sciences) or using an Odyssey CLx scanner (LiCor) for fluorescence signaling. Blots were quantified using ImageJ 1.52 k.

### Luciferase reporter assay

The expression of inducible luciferase in cell lines was measured using the Dual-Glo Luciferase Assay system (cat. no. E2920, Promega). Cells were seeded at a density of 15,000 cells per well in a 96-well plate (3917, Costar). The following day, 500 ng/mL dox (Takara Bio) was added to the media, and cells were incubated for 48 h at 37 °C. Luciferase buffer and substrate reagents were prepared per the manufacturer's instructions. Briefly, the Dual-Glo Luciferase reagent was added to a 1:1 ratio to the cells and allowed to incubate at room temperature (25 °C) for 10 min in the darkness on an orbital shaker. Firefly luciferase expression (luminescence) was measured using the SpectraMax M5 plate reader and Dual-Glo luciferase protocol with an integration time of 500 ms.

### Growth curves

Cells were seeded in 6 technical replicates at a density of 5000 cells per well in a 96-well plate. The next day, cells were treated with 0–500 ng/mL dox (Takara Bio) and imaged in an Incucyte Live-Cell Analysis system (Sartorius) to track confluency over the course of 7 days. Images were acquired with a 10X objective every 6 h, and analyzed with custom-defined masks to identify the area of each cell line. Media was replaced with fresh dox every 48 hours.

### Nascent protein synthesis assay

Nascent protein synthesis was measured with the Click-iT HPG Alexa Fluor 488 Protein Synthesis Assay Kit (cat. no. C10428, ThermoFisher). Cells were plated in 12-well plates (Cell Treat) containing 18 mm cover glasses (Fisher Scientific) at the bottom, and treated with 500 ng/mL of dox (Takara Bio) for 48 h. Following induction, cells were washed with PBS and incubated with 50 M of Click-iT homopropargylglycine (HPG) in methionine-free RPMI media (Gibco) for 30 min. Next, cells were fixed with 3.7% formaldehyde in PBS and permeabilized using 0.5% Triton X-100. The Click-iT reaction was performed per manufacturer's protocol by preparing a reaction cocktail mix containing Alexa Fluor 488 azide. DNA staining was performed using NuclearMask Blue Stain, and incubated for 30 min in the dark. Coverslips were then removed from the plate and mounted on glass slides using a Slow-Fade mounting solution (Fisher Scientific). Slides were imaged in an

Olympus IX73 inverted microscope using the CellSens Software (Olympus). GFP signal intensity was quantified and normalized to the total number of cells using Image J.

## Cell cycle analysis by flow-cytometry

Cell cycle analysis was performed by in vitro labeling of cells with the APC BrdU Flow Kit (cat. no. 552598, BD Biosciences). First, 1 million cells were treated with 500 ng/mL dox for 48 h. Next, 10 μM BrdU solution was added to the culture media, and cells were incubated for 2 h. Cells were then fixed and permeabilized according to the manufacturer's protocol. Next, samples were treated with 60 μg of DNase and incubated for 1 h at 37 °C to expose the incorporated BrdU. Finally, cells were stained for BrdU (1:50 antibody dilution) and total DNA (7-AAD 1:50 solution dilution) and analyzed by flow cytometer in a BD LSRFortessa cell analyzer (BD Biosciences) using 488 nm and 640 nm lasers. Cells were gated and assigned to the G1, S, or G2 phases of the cell cycle.

## Quantification of apoptosis by flow-cytometry

Apoptotic cells were labeled for flow cytometry using a Dead Cell Apoptosis Kit with Annexin V FITC & Propidium Iodide (cat. no. V13242, Invitrogen). Briefly, cells were seeded at a density of 500,000 cells/well in a 6-well plate and treated with 500 ng/mL dox for 72 h. Next, cell pellets were collected and resuspended in 100 μL of 1X annexin-binding buffer containing 5 μL of FITC annexin V antibody and 1 μL of 100 μg/mL propidium iodide (PI) working solution. Following an incubation at room temperature for 15 min, 400 μL of 1X annexin-binding buffer was added to each sample. Stained cells were analyzed by flow cytometry in a BD LSRFortessa cell analyzer (BD Biosciences), measuring the fluorescence emission at 530 nm (FITC) and 575 nm (PI). Cells were gated and classified as live (negative for both annexin V and PI), early apoptotic (positive for annexin V), or late apoptotic (positive for both annexin V and PI). Image quantification was performed in FlowJo v 10.8.1 software.

## Immunofluorescence analysis

Cells were seeded at a density of 40,000 cells per well in a 12-well plate with round coverslips. After 48 h treatment with 500 ng/mL dox, cells were subjected to 2 Gy X-ray ionizing radiation (IR). At each time point (0, 15, 30, 60, 120, and 360 min post-radiation), cells were washed with CSK buffer for 5 min and later incubated with 0.7% Triton X-100 in CSK for 5 min on ice. Fixation was done with 4% paraformaldehyde in CSK at room temperature for 30 min. After two washes with PBS, cells were permeabilized with 0.2% Triton X-100 in PBS and blocked with PBS-T containing 5% BSA for 1 h. Incubation with 1:3000 or 1:100 dilution primary antibodies against mouse γH2A.X (cat. no. 05-636, Millipore), 53BP1 (cat. no. NBP2-25028, Bio-Techne), RAD51 (cat. no. sc-398587, Santa Cruz Biotechnology), or STING (cat. no. PA5-23381, Invitrogen) was done overnight at 4 °C or 1 h for the last 2 antibodies. After 3 washes with PBS, cells were incubated with fluorescent AF-568 (cat. no A11031, Invitrogen) or AF-647 (cat. no. A21236, Invitrogen) goat anti-mouse secondary antibodies and Hoechst 33342 nucleic acid stain (cat. no. H3570, Thermo Fisher) at a 1:10,000 dilution for 1 h in the dark. Coverslips were mounted on glass slides by addition of SlowFade Diamond antifade mountant solution (cat. no. S36963, ThermoFisher), and visualized either in an Olympus IX73 Inverted Microscope (γH2A.X immunofluorescence) or in an SP8 TCS Leica confocal microscope (53BP1, RAD51, and STING immunofluorescence). Image processing, quantification of γH2AX, 53BP1, and RAD51 foci per nucleus, as well as quantification of STING area, were performed in ImageJ2 v 2.9.0.

## Double-strand-break repair GFP reporters

DNA repair pathway choice was measured in U2OS EJ7-GFP (c-NHEJ) and U2OS DR-GFP (HR) cells[57] by flow cytometry. Cells were seeded at a density of 500,000 cells/well in 12-well plates. The next day, cells were

transfected with 500 ng pCAGGS-NZEGFP (transfection efficiency control), 500 ng pCAGGS-BSKX (empty vector control), 500 ng pCBASce (ISceI; for HR assay) and 250 ng pX330-7a/ 250 ng pX330-7b ('7a' and '7b' guide RNAs; for c-NHEJ assay) plasmids using the Lipofectamine 3000 Transfection Reagent (cat. no. L3000001, Invitrogen). After 8 h, fresh media was added with 500 ng/mL dox. After 48 h, cells were collected, resuspended in 500 μL eBioscience flow cytometry staining buffer (cat. no. 00-4222-26, Invitrogen) and analyzed in a BD LSRFortessa cell analyzer (BD Biosciences), measuring the fluorescence emission at 530 nm (GFP). GFP-positive cells were quantified in each condition with FlowJo v 10.8.1 software.

## Drug dose-response curves

Cells were seeded at a density of 10,000 cells per well in a 96-well plate and treated with 500 ng/mL dox. The next day, cells were treated with 0–100 μM of the DNA-PK inhibitor NU7441 (cat. no. HY-11006, MedChemExpress) or RAD51 inhibitor B02 (cat. no. HY-101462, MedChemExpress) and incubated for 72 h. The CellTiter-Glo 2.0 viability assay (cat. no. G9241, Promega) was used to measure the level of ATP as a surrogate for cell viability. Luminescence was measured in a SpectraMax M5 plate reader (Associated Technologies Group) with the SoftMax Pro software using the CellTiter-Glo protocol with an integration time of 500 ms. For determining response to STAT3 inhibition, cells were instead treated with the STAT3 inhibitors C188-9 (cat. no. 30928, Cayman Chemical) and HJC052 (cat. no. 22351, Cayman Chemical), and incubated for 7 days. To determine cell viability, 20 μL of MTT solution (5 mg/mL) was added to each well in the 96-well plate. Cells were incubated for 2–4 h at 37 °C to allow MTT to be metabolized. After incubation, the MTT-containing medium was carefully removed, and 50 μL of DMSO was added to each well to dissolve the formazan crystals. Plates were incubated and gently shaken for 5 min to ensure complete solubilization. The absorbance was then measured at 570 nm using a microplate reader.

## Time-lapse imaging of mitosis

PIGPZ–H2B–Cherry was subcloned from an existing H2B–Cherry construct using NotI and BsrGI sites. Lentiviral transduction of pIGZ H2B–Cherry was achieved by transfecting 3 μg of the desired plasmid supplemented with the following packaging plasmids: 3 μg pSPAX2 and 1 μg pMD2.G into HEK-293T cells. Plasmids psPAX2 (Addgene plasmid, 12260) and pMD2.G (Addgene plasmid, 12259) were gifts from Prof. D. Trono (École Polytechnique Fédérale de Lausanne). After 48 h, the medium was collected, passed through a 0.45 μm filter (VWR Science), and transferred onto the target cells. To increase transduction efficiency, polybrene (Millipore) was added to a final concentration of 8 μg/ mL. For time-lapse imaging, 60,000 cells expressing H2B–mCherry were first seeded per well on imaging chambers (Lab-Tek) treated with 2 Gy IR, and immediately replaced with fresh medium. All cells were subsequently imaged for 8 h on a DeltaVision Elite imaging station (GE Healthcare), equipped with a CoolSNAP HQ2 camera, a 40 × 0.6 NA immersion objective (Olympus), and DeltaVision softWoRx software. Images were acquired at 6 min intervals and included z-stacks of 20 images at 0.4 μm intervals. Image analysis was done using ICY software (Institut Pasteur). Only cells that entered mitosis and stayed in the frame throughout the imaging session were included for analysis.

## Co-culture with NK-92 cells

Cells were plated at a density of 20,000 cells per well in a 24-well plate in RPMI containing 500 ng/mL dox. After 48 h, cell lines were exposed to 2 Gy irradiation (Cesium-137 γ-ray, IBL 637). NK-92 cells (cat. no. CRL-2407, ATCC) were plated at a ratio of 10:1 on cells and left for 48 h in NK-92 media (12.5% FBS, 12.5% HS, 50 μM, β-mercaptoethanol, 1% PenStrep, and RPMI 1640), containing 100 Iμ/mL Recombinant IL-2 (200-02-50UG, PeproTech) and 500 ng/mL doxycycline. Cells were then washed twice

with PBS before the addition of 4% PFA for 10 min. Cells were then washed twice with PBS before adding 200 µL Crystal Violet solution (cat. no. C3886, Sigma) for 5 min with gentle rocking. Cells were then washed with dH2O three times before drying overnight. To quantify the cell coverage within each well, plates were imaged in a BioRad Chemidoc MP imaging system and analyzed using the 'ColonyArea' Fiji plugin[74]. Colony coverage was compared to "Control #2", where cancer cells were grown in NK-92 media but in the absence of NK-92 cells.

## Reporting summary

Further information on research design is available in the Nature Portfolio Reporting Summary linked to this article.

## Data availability

CCLE data was downloaded from DepMap (https://depmap.org/portal/data_page/?tab=allData)[75], including cell line annotations ("Cell_lines_annotations_20181226.txt"), RNA-seq gene counts ("CCLE_RNAseq_genes_counts_20180929.gct.gz"), $\log_2$ copy number changes over the mean per gene ("CCLE_copynumber_byGene_2013-12-03.txt.gz"), and WGD ("OmicsSignatures.csv"). TCGA copy number call significance was obtained by GISTIC copy number calls from the Broad TCGA copy number portal (tumorscape.org) using the Tumorscape 1.2.1 analysis data ("2015-06-01 stddata_2015_04_02 arm-level peel-off", available from the original authors on request)[1,76]. Tumor purity estimates were used from the ESTIMATE algorithm[77] applied to the extended TCGA cohorts (https://static-content.springer.com/esm/art%3A10.1038%2Fncomms9971/MediaObjects/41467_2015_BFncomms9971_MOESM1236_ESM.xlsx)[78]. Oncogenes and Tumor Suppressor gene lists were downloaded from the COSMIC gene census ("Census_COSMIC96.tsv", requires an account at https://cancer.sanger.ac.uk/)[38] and included in the respective lists if they were listed as Hallmark, Tier 1, or Tier 2. Individual copy number events were extracted from previously published Ziggurat Deconstruction analysis of copy number states ("TCGA.all_cancers.150601.zigg_events.160923.txt", available from the original authors on request)[76]. WGD information was downloaded from the TCGA Pan-Cancer Atlas (https://api.gdc.cancer.gov/data/4f277128-f793-4354-a13d-30cc7fe9f6b5). For patient survival analyses, we used recently published data from a colorectal cancer cohort ("Supplementary_Table_01.xlsx" at https://static-content.springer.com/esm/art%3A10.1038%2Fs41586-024-07769-3/MediaObjects/41586_2024_7769_MOESM3_ESM.zip)[64]. Source data are provided in this paper.

## Code availability

Unless stated otherwise, computational analyses have been performed using R 4.3.1. Workflows were orchestrated with *Snakemake* (7.32.4)[79] and parallelized with *clustermq* (0.8.915)[80]. All analysis code, including used packages and their versions, is available at https://github.com/mschubert/ToxicGenes under the GNU General Public License version 3.

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

## Acknowledgements

We thank the following investigators for providing data related to ORF screens: Cory Johannessen and Tikvah K. Hayes in Meljuso, OVSAHO, and Kuramochi cell lines; Nikhil Wagle, Pingping Mao, and Ofir Cohen for T47D cells; Adam Bass, Bruno Bockorny and Maria Rusan for NCI-H2077 cells. U2OS reporter cell lines were kindly provided by the Stark Laboratory and used in the analysis of DNA damage repair as described in ref. 57. This work was funded by a Marie Skłodowska-Curie post-doctoral fellowship (101068734) to MSR, a Dutch Cancer Society grant (18-RUG-11457) to F.F., and support from The Brain Tumor Charity and The Swedish Research Council (2022-01539) to V.R. Work in the U.B.-D. lab was supported by an ERC Starting Grant (StG #945674). K.S. was supported by NIH 1R35 CA283977-01. R.B. was supported by The Gray Matters Brain Cancer Foundation, The Brain Tumor Charity, Break Through Cancer, and the Brown Fund for Innovative Cancer Informatics.

## Author contributions

U.B-D., R.B., and F.F. directed the project. M.S and V.R. collected data and performed computational analyses. V.R., N.K., M.F.S.P.R., D.W., X.L., A.v.d.B., and J.R. carried out experimental work, where N.K. and M.F.S.P.R. contributed equally. K.H., M.Sw., Y.H., and D.C. assisted with the evaluation of DNA damage response. R.S. conducted image analysis from immunofluorescence data. J.Z. and U.B-D. provided RPE-1 cell line data. P.B., L.M.G, J.H.H., A.I., S.M., J-H.S., E.S., L.G., W.C.H., and K.S. provided ORF screen data. R.H.M and M.C.T. aided with result interpretation. V.R., M.S., U.B-D., R.B., and F.F. wrote the manuscript.

## Funding

## Competing interests

K.S. is on the SAB and has stock options with Auron Therapeutics. K.S. receives grant funding from Novartis and KronosBio on topics unrelated to this manuscript. W.C.H. is a consultant for Thermo Fischer Scientific, Solasta Ventures, MPM capital, KSQ Therapeutics, Tyra Biosciences, Frontier Medicines, Jubilant Therapeutics, RAPPTA Therapeutics, Serinus Biosciences, Hexagon Biosciences, Kestral Therapeutics, Function Oncology, and Calyx. R.B. consults for Scorpion Therapeutics and receives grant funding from Novartis. U.B.D. consults for Accenet Therapeutics and received funding from Novocure. F.F. is co-founder and Chief Scientific Officer of iPsomics. The work in this study is unrelated to this role. The remaining authors declare no competing interests.

## Additional information

[1]Department of Medical Oncology and Center for Neuro-Oncology, Dana-Farber Cancer Institute, Boston, MA, USA. [2]Department of Cancer Biology, Dana-Farber Cancer Institute, Boston, MA, USA. [3]Harvard Medical School, Boston, MA, USA. [4]Broad Institute of Harvard and MIT, Cambridge, MA, USA. [5]Department of Immunology, Genetics and Pathology, Uppsala University, Uppsala, Sweden. [6]Oncode Institute, Division of Cell Biology, The Netherlands Cancer Institute, Amsterdam, Netherlands. [7]European Research Institute for the Biology of Ageing, University Medical Center Groningen, Groningen, Netherlands. [8]Institute of Computational Biology, Helmholtz Munich, Neuherberg, Germany. [9]Institute of Bioinformatics, Medical University of Innsbruck, Innsbruck, Austria. [10]Department of Radiation Oncology, Dana-Farber Cancer Institute, Boston, MA, USA. [11]Department of Human Molecular Genetics & Biochemistry, Faculty of Medicine, Tel Aviv University, Tel Aviv, Israel. [12]Department of Pediatrics, Dana-Farber Cancer Institute, Boston, MA, USA. [13]St. Jude Children's Research Hospital, Department of Oncology, Memphis, TN, USA. [14]Division of Hematology, Oncology, and Transplantation, University of Minnesota, Minneapolis, MN, USA. [15]Department of Cancer Biology, Perelman School of Medicine at the University of Pennsylvania, Philadelphia, PA, USA. [16]Biomedical Center (BMC), Physiological Chemistry, Ludwig Maximilians University, Munich, Germany. [17]These authors contributed equally: Veronica Rendo, Michael Schubert. [18]These authors jointly supervised this work: Uri Ben-David, Rameen Beroukhim, Floris Foijer. ✉e-mail: veronica.rendo@igp.uu.se; m.schubert@nki.nl; ubendavid@tauex.tau.ac.il; rameen_beroukhim@dfci.harvard.edu; f.foijer@umcg.nl

