## [Peer Review File · Nature Communications]

A compendium of Amplification-Related Gain Of Sensitivity (ARGOS) genes in human cancer

Corresponding Author: Dr Veronica Rendo

Editorial Note: This manuscript has been previously reviewed at another journal that is not operating a transparent peer review scheme. This document only contains reviewer comments and rebuttal letters for versions considered at *Nature Communications*. Mentions of the other journal have been redacted.

Version 0:

Reviewer comments:

Reviewer #1

(Remarks to the Author)

The authors have done a good job addressing the concerns raised during the first round of review. By toning down some of the language regarding novelty and placing their work in a broader context, I believe that the paper will soon be appropriate for publication.

I have a few remaining questions -

The authors write, "Many of the publications referenced by the Reviewer focus on protein compensation. However, compensation at the transcriptome level is less well-studied. Multiple paths exist to adjust protein levels based upon their expression and activity, and these prevent overexpression of many genes from being toxic. However, we see modulation of transcriptomic levels as a more direct indication of gene toxicity. Notably, proteomic compensation does not predict transcriptomic compensation (Reviewer Fig 1). For that reason, we focus on transcriptomic compensation here." This raises an interesting topic. In my view, the effects of CNVs on non-coding RNAs, microRNAs, lncRNAs, etc are very poorly understood. If the authors seek to differentiate this work by focusing on transcriptional compensation, could they also include a small analysis of non-coding RNA expression? Does any toxicity data exist on ncRNA expression. Please note that I do not consider this line of investigation essential for publication, particularly if such datasets do not exist.

The finding that RBM14 overexpression is associated with an improved response to radiation is interesting. In light of the recent Bei Cancer Discovery paper, I think that the authors should discuss more the potential therapeutic implications of toxic overexpression and how it could be taken advantage of.

Does a cell's basal ploidy affect the ARGOS genes? If a cell has undergone whole-genome doubling, I could imagine that that confers greater tolerance to gene expression. Have there been overexpression screens conducted in WGD+ cell lines?

Reviewer #2

(Remarks to the Author)

I reviewed this paper previously for [Redacted] (I was broadly positive about the work!) and so my comments here relate to the author's responses to my comments and the (slightly) revised version of the manuscript.

Strengths

Comprehensive Dataset - The study leverages extensive datasets from TCGA and CCLE, enhancing the robustness of the findings.

Novelty - The concept of ARGOS genes provides a fresh perspective on cancer gene amplifications and their impact on cellular fitness.

Potential Therapeutic Targets - Identifying ARGOS genes opens new avenues for therapeutic interventions in cancers with specific gene amplifications.

Weaknesses

Limited Experimental Validation - Only two ARGOS genes (CDKN1A and RBM14) were experimentally validated. Further validation of additional ARGOS genes would strengthen the overall conclusions but i appreciate that the authors have done enough for this paper.

Mechanistic Insights - While the study suggests potential mechanisms of toxicity for RBM14, the exact pathways and interactions remain to be fully elucidated.

I have several questions for the authors

Mechanistic Insights on RBM14: Can you provide more detailed mechanistic insights into how RBM14 overexpression leads to altered DNA damage response and innate immune signaling? Are there specific pathways or interactions you are focusing on?

In Vivo Models: Do you plan to extend your validation to in vivo models to confirm the effects of ARGOS gene overexpression observed in vitro?

Therapeutic Implications: Based on your findings, how do you envision the therapeutic exploitation of ARGOS genes? Are there specific cancer types or patient populations that might benefit most from targeting these genes?

Data Sharing and Collaboration: how will you make the comprehensive catalog of compensated, toxic, and ARGOS genes available to the research community? How do you plan to facilitate further research and collaboration in this area? As before i really think this is key for this paper and i can't see it in the paper.

Reviewer #3

(Remarks to the Author)

My concerns have largely been addressed and/or will be addressed in the proposed set of experiments which should make the manuscript suitable for publication in Nature Communications.

Version 1:

Reviewer comments:

Reviewer #1

(Remarks to the Author)

My concerns have been addressed, and I support the publication of this work. I congratulate the authors for their discoveries.

Reviewer #2

(Remarks to the Author)

In general the authors have responded to my comments well. I do, however, wonder how they expect readers to explore their data? I had previously suggested some sort of a browser or including it with the DepMap data. I really think this is important other readers of the paper will not be able to explore this study or glean their own insights.

Reviewer Comments

Reviewer #1 (Remarks to the Author):

The authors have done a good job addressing the concerns raised during the first round of review. By toning down some of the language regarding novelty and placing their work in a broader context, I believe that the paper will soon be appropriate for publication.

I have a few remaining questions -

The authors write, "Many of the publications referenced by the Reviewer focus on protein compensation. However, compensation at the transcriptome level is less well-studied. Multiple paths exist to adjust protein levels based upon their expression and activity, and these prevent overexpression of many genes from being toxic. However, we see modulation of transcriptomic levels as a more direct indication of gene toxicity. Notably, proteomic compensation does not predict transcriptomic compensation (Reviewer Fig 1). For that reason, we focus on transcriptomic compensation here." **This raises an interesting topic. In my view, the effects of CNVs on non-coding RNAs, microRNAs, lncRNAs, etc are very poorly understood. If the authors seek to differentiate this work by focusing on transcriptional compensation, could they also include a small analysis of non-coding RNA expression? Does any toxicity data exist on ncRNA expression. Please note that I do not consider this line of investigation essential for publication, particularly if such datasets do not exist.**

We thank the reviewer for their comment. We have now compared how often protein-coding genes occur in our compensated genes compared to non-coding genes (lncRNA, pseudogenes). We find that protein-coding genes are compensated more often (**Reviewer Figure 1**), and have added these results to Fig. S2e. We have not identified a suitable set of ORF screens for non-coding gene toxicity. Note that RRM/PLD comparisons are included in this panel for the previous revision round.

Reviewer Figure 1. Changes between expected and observed compensation frequency for different classes of genes. Shown are protein-coding genes (more frequently compensated), lncRNAs, and pseudogenes (both less frequently compensated). RNA recognition motif (RRM) and Prion-Like Domain (PLD) differences are from previous revision round.

The finding that RBM14 overexpression is associated with an improved response to radiation is interesting. In light of the recent Bei Cancer Discovery paper, I think that the authors should discuss more the potential therapeutic implications of toxic overexpression and how it could be taken advantage of.

We thank the Reviewer for pointing out this reference. We have now included a sentence on how our work is complementary to previously identified genetic passenger dependencies in the study published by Bei et al. 2024.

Does a cell's basal ploidy affect the ARGOS genes? If a cell has undergone whole-genome doubling, I could imagine that that confers greater tolerance to gene expression. Have there been overexpression screens conducted in WGD+ cell lines?

We have repeated our compensation analysis in both CCLE and TCGA data splitting the cell lines and tumors by their WGD status according to DepMap. We have also repeated the Toxicity analysis in the same manner. The results of this analysis are shown below (**Reviewer Figure 2**) and have been added in Fig. S4d-f.

We find no strong general effect in terms of changed levels of compensation or toxicity. There are, however, individual genes that show distinct patterns, especially in CCLE compensation. We do, however, not rule out that this was caused by the reduced statistical power from a lower sample size.

Reviewer Figure 2. Effect of WGD+ and WGD- samples in compensation for CCLE and TCGA, as well as ORF overexpression toxicity.

Reviewer #2 (Remarks to the Author):

I reviewed this paper previously for [Redacted] (I was broadly positive about the work!) and so my comments here relate to the author's responses to my comments and the (slightly) revised version of the manuscript.

Strengths

Comprehensive Dataset - The study leverages extensive datasets from TCGA and CCLE, enhancing the robustness of the findings.

Novelty - The concept of ARGOS genes provides a fresh perspective on cancer gene amplifications and their impact on cellular fitness.

Potential Therapeutic Targets - Identifying ARGOS genes opens new avenues for therapeutic interventions in cancers with specific gene amplifications.

Weaknesses

Limited Experimental Validation - Only two ARGOS genes (CDKN1A and RBM14) were experimentally validated. Further validation of additional ARGOS genes would strengthen the overall conclusions but I appreciate that the authors have done enough for this paper.

Mechanistic Insights - While the study suggests potential mechanisms of toxicity for RBM14, the exact pathways and interactions remain to be fully elucidated.

I have several questions for the authors

Mechanistic Insights on RBM14: Can you provide more detailed mechanistic insights into how RBM14 overexpression leads to altered DNA damage response and innate immune signaling? Are there specific pathways or interactions you are focusing on?

In the original manuscript, we describe a link between RBM14 overexpression, altered DNA damage response, and innate immune signaling in lung and breast cancer cell lines. Specifically, we studied how RBM14 overexpression regulates: i) DNA damage repair (by immunofluorescent-based quantification of γ H2AX foci in irradiated cells), ii) rates of homologous recombination vs non homologous end joining-mediated repair (in U2OS cells with repair-specific GFP reporters), and iii) STING activation (by immunofluorescence-based quantification of perinuclear STING area). We have now conducted several experiments that provide more mechanistic insights into these cellular phenotypes. Regarding RBM14's role in modulating DNA repair pathway choices, we have now performed orthogonal assays that confirm how RBM14 overexpression leads to increases in c-NHEJ-mediated DNA repair while decreasing HR-mediated repair (please see the response to Reviewer 3 below). Regarding RBM14's role in modulating innate immune signaling, we have specifically looked deeper into STING pathway activation, and how the STING-STAT3 axis potentially induces NK-mediated tumor cell killing.

First, we looked into how RBM14 overexpression may modulate downstream STING signaling in our cell line models. We have recently shown how inflammatory responses in chromosomally unstable breast cancers are accompanied by increased STAT3 signaling, which is in itself dependent on STING pathway activation¹ and has been linked to immunosuppressive phenotypes². To determine whether RBM14 affects the STING/STAT3 signaling axis, we evaluated the effect of gene overexpression on cellular response to STAT3 inhibitors. Briefly, we induced DNA damage in our set of RBM14- vs. luciferase-overexpressing lung and breast cancer cell lines, and treated with the STAT3 inhibitors C188-9 and HJC052 for 72 h. We found a decreased sensitivity of RBM14-overexpressing cells when treated with 1 μ M of the STAT3 inhibitors C188-9 and HJC052 compared to luciferase-expressing control cells (**Reviewer Figure 3**). However, we do note that RBM14-overexpressing cells had a starting lower viability when compared to luciferase control cells (**Reviewer Figure 4**). These results suggest a potential role for RBM14 in impairing STAT3 signaling, thus reducing the effects of STAT3 inhibitors and contributing to an inflammatory phenotype in our cell line models. The latter could well be the result of increased STAT1 activity¹. We have added the STAT3 inhibition data as Figure 7c on the revised manuscript.

Reviewer Figure 3. Effect of STAT3 inhibition in RBM14-overexpressing cells (blue) compared to luciferase controls (red). Cell viability is shown for cells treated with 1 μ M of either C188-9 (top) or HJC052 (bottom) STAT3 inhibitors relative to DMSO control, both in the absence (circled symbol; solid line) or presence (squared symbols; dashed line) of 2 Gy DNA damage-inducing irradiation (IR).

Reviewer Figure 4. Drug response curves for RBM14- vs luciferase-overexpressing cells following treatment with the STAT3 inhibitors (a) C188-9 and (b) HJC052 for 72 h. Cell viability is quantified relative to luciferase control. Conditions including 2 Gy irradiation are shown in dashed lines.

Next, we sought to confirm whether RBM14 overexpression increases the rates at which the immune system -in particular natural killer (NK) cells- recognizes and kills cancer cells with gene overexpression. To this end, we irradiated RBM14- and luciferase-overexpressing cancer cells and 2 h later added NK-92 cells to the culture at 10:1 ratios. After 48 h of co-culture, we removed the NK-92 cells and fixed and stained the cancer cells with crystal violet, and quantified the area of colony formation for RBM14-overexpressing cancer cell lines relative to their luciferase controls (**Reviewer Figure 5**). Quantification of colony area was internally normalized to a culture condition with cancer cell lines growing in the absence of NK-92 cells (“control #2” in **Reviewer Figure 5**).

A decrease in colony area was observed in both lung NCI-H838^{RBM14} and NCI-H1650^{RBM14} cells, with the largest effect size (> 60% reduction in relative colony area; $P = 0.0192$) being present in HCC70^{RBM14} cells. This experiment confirms that RBM14 overexpression results in increased immune recognition and NK cell-mediated tumor death. We have added these results as Figure 7d on the revised manuscript.

Reviewer Figure 5. Determination of NK-mediated tumor cell killing by co-culture of RBM14- and luciferase-overexpressing cells with NK-92 cells after 2 Gy irradiation. Representative images of each cell line and culturing condition. Colony area was quantified after 48 h of co-culture relative to “control #2” condition (only cancer cells, grown in NK-92 cell media). Colony area is represented for each RBM14-overexpressing cell line relative to its corresponding luciferase control.

Therapeutic Implications: Based on your findings, how do you envision the therapeutic exploitation of ARGOS genes? Are there specific cancer types or patient populations that might benefit most from targeting these genes?

We agree with the Reviewer that showing therapeutic implications for ARGOS genes, or in particular RBM14, would strengthen our point about collateral sensitivities. We now include new data from a clinical colorectal cancer cohort, where patients have been treated with standard-of-care regimens of radiotherapy³. Here, we show that when *CCND1* is amplified without *RBM14* it is associated with worse response to radiation treatment, but when *RBM14* (which resides next to it) is also amplified, the patients respond significantly better to the treatment (Reviewer Figure 6; $P = 0.048$, Cox model). In these data, RBM14 is indeed partially compensated. This is in line with our *in vitro* demonstration that RBM14 overexpression causes an altered DNA damage response. These findings represent a proof-of-concept that copy number states of ARGOS genes can serve as biomarkers of treatment response. We have added these new results as Figure 8a-c in the revised manuscript and commented on the potential therapeutic implications for this ARGOS gene in the discussion.

Reviewer Figure 6. Impact of CCND1/RBM14 amplifications on patient survival in a CRC cohort. (a) Copy number and gene expression of CCND1 and (b) RBM14. (c) Overall patient survival by co-amplification of CCND1 and RBM14 vs. only CCND1. Cox Proportional Hazards model adjusting for sex and age $P=0.048$.

In Vivo Models: Do you plan to extend your validation to *in vivo* models to confirm the effects of ARGOS gene overexpression observed *in vitro*?

The validation of RBM14’s mechanism of toxicity *in vivo* is beyond the scope of our current work and the timeline for revision (4 weeks + 4 week extension), but we recognize the importance of validating the altered DNA damage response and immune signaling phenotypes in mouse xenografts and other

relevant models. In an effort to extend the translatability of our *in vitro* results, we have now included the exciting clinical data shown above (where gains of RBM14 are associated with altered sensitivity to radiation in a cohort of colorectal cancer patients³). We believe this constitutes strong evidence for the clinical relevance of our findings.

Data Sharing and Collaboration: how will you make the comprehensive catalog of compensated, toxic, and ARGOS genes available to the research community? How do you plan to facilitate further research and collaboration in this area? As before i really think this is key for this paper and i can't see it in the paper.

We apologize that we have not indicated where to find the data of our results, which we explicitly stated we provide. The results of both the Compensation as well as the Toxicity analyses are provided as Supplementary Tables, which may not have been visible enough for the review process. We have now added explicit references to the corresponding Supplementary Tables when mentioning this in-text. We initially planned on integrating our results with the DepMap analysis platform, but this was unfortunately not possible as it relies on cell line-level measures and our Compensation and Toxicity scores are gene-level measures. We unfortunately also consider developing our own interactive exploration platform out of scope for this publication. However, we will gladly consider developing a Shiny app for this in a follow-up project.

Reviewer #3 (Remarks to the Author):

My concerns have largely been addressed and/or will be addressed in the proposed set of experiments which should make the manuscript suitable for publication in Nature Communications.

Reviewer 3 previously requested a more thorough characterization of RBM14's overexpression effects. In our response to Reviewer 2, we provide novel data that expands our understanding of how RBM14 overexpression leads to altered innate immune signaling following DNA damage, in particular: i) the upregulation of cytokines (IL-6, IL-8) and chemokines (CXCL10 and CCL5) as a result of STING pathway activation, and ii) the increase in cancer cell recognition and death by natural killer cells in co-culture experiments involving RBM14-overexpressing cancer cells and NK-92 immune cells (see detailed response above).

In our original manuscript, we additionally report changes in DNA repair pathway choices following RBM14 overexpression. Reviewer 3 pointed out that the quantification of DNA-PKcs protein levels in RBM14 vs. luciferase overexpressing cells should be conducted in samples belonging to the same immunoblot. We have now redone this experiment and quantified relative levels of activated protein (measured as phosphorylation of the S2056 residue). In NCI-H838 and NCI-H1650 lung cancer cell lines, we observed that RBM14 overexpression is sufficient to activate pDNA-PKcs protein levels by ≥ 2.6 -fold following 2 Gy irradiation. In breast cancer cell lines, ZR-75-1 showed the largest effect size in protein upregulation. **(Reviewer Figure 8)**. We have updated Figure 6b of the original manuscript with the revised immunoblots and quantifications (new Figure 6d).

Reviewer Figure 8. Levels of phosphorylated DNA-PKcs (Ser2056) in lung NCI-H838 and NCI-H1650 cells 15 min after 2 Gy irradiation. Protein expression in RBM14-overexpressing cells was quantified relative to luciferase control.

To examine DNA repair pathway decisions further, we have now included immunofluorescence-based quantifications of RAD51 [marker of repair by homologous recombination (HR); data missing in original manuscript] and 53BP1 [marker of repair by non-homologous end joining (NHEJ); newly generated data] foci after irradiation. RAD51 foci quantification at 120 min following 2 Gy irradiation showed a decrease in foci for 3 out of 4 RBM14-overexpressing cell lines when compared to luciferase controls, suggesting a decrease in HR-mediated DNA repair (**Reviewer Figure 9**). These results are in line with the U2OS GFP reporter data shown in **Figure 6c** of the original manuscript.

Reviewer Figure 9. Quantification of RAD51 foci in RBM14- and luciferase-overexpressing cells 120 min following DNA damage-inducing irradiation. Representative images are shown for each cell line and condition. Scale bar: 20 μ m

In contrast, we observed a significant increase in 53BP1 foci at 60 min when inducing DNA damage with 2 Gy in NCI-H838^{RBM14} and ZR-75-1^{RBM14} cells by 2 Gy irradiation. This trend was also present in HCC70^{RBM14} but not NCI-H1650^{RBM14} cells (**Reviewer Figure 10**). Overall, this is supportive of our orthogonal approaches implicating RBM14 in c-NHEJ-mediated DNA repair.

Reviewer Figure 10. Quantification of 53BP1 foci in RBM14- and luciferase-overexpressing cells 60 min following DNA damage-inducing irradiation. Representative images are shown for each cell line and condition. Scale bar: 20 μm

Altogether, our analyses suggest a role for RBM14 in mediating DNA repair choices, with increased rates of NHEJ-mediated repair and decreased rates of HR-mediated repair following gene overexpression. We have added these new supporting results as Figures 6b-c in the revised manuscript.

References

1. Hong, C. *et al.* cGAS-STING drives the IL-6-dependent survival of chromosomally unstable cancers. *Nature* **607**, 366–373 (2022).
2. Hillmer, E. J., Zhang, H., Li, H. S. & Watowich, S. S. STAT3 signaling in immunity. *Cytokine Growth Factor Rev.* **31**, 1–15 (2016).
3. Nunes, L. *et al.* Prognostic genome and transcriptome signatures in colorectal cancers. *Nature* **633**, 137–146 (2024).

Response to Reviewer Comments

Below, we outline our responses (in blue text) to each point raised by the Reviewers (in bold).

Reviewer #1 (Remarks to the Author):

My concerns have been addressed, and I support the publication of this work. I congratulate the authors for their discoveries.

We thank Reviewer #1 for their comments, which improved the final version of our manuscript.

Reviewer #2 (Remarks to the Author):

In general the authors have responded to my comments well. I do, however, wonder how they expect readers to explore their data? I had previously suggested some sort of a browser or including it with the DepMap data. I really think this is important other readers of the paper will not be able to explore this study or glean their own insights.

We thank Reviewer #2 for this comment, and we recognize the importance of making our data easily explorable. To address this, we have taken two steps: first, we have added the results of our compensation and toxicity analyses, both across cancers and for individual cancer types, as Supplementary Data files. Following our provided guidelines, a reader can explore this data (using e.g. Microsoft Excel or GraphPad Prism) and determine the compensation and toxicity score of their gene(s) of interest. The reader can also determine the context (e.g. cancer type specificity) of an ARGOS gene of interest. Second, we have specifically created custom gene set files for our set of compensated, toxic, and ARGOS genes that are compatible for upload in the Enrichr platform and for use as custom signatures when performing Gene Set Enrichment Analysis (GSEA). We believe these new files (available as Supplementary Information) will facilitate follow-up exploration and analysis of our data.